# Surface temperature response to the major volcanic eruptions in multiple reanalysis data sets

Masatomo Fujiwara[1], Patrick Martineau[2], Jonathon S. Wright[3]

[1]Faculty of Environmental Earth Science, Hokkaido University, Sapporo, 060-0810, Japan
[2]Research Center for Advanced Science and Technology, The University of Tokyo, Tokyo, 153-8904, Japan
[3]Department of Earth System Science, Tsinghua University, Beijing, 100084, China

*Correspondence to*: Masatomo Fujiwara (fuji@ees.hokudai.ac.jp)

**Abstract.** The global response of air temperature at 2 metre above the surface to the eruptions of Mount Agung in March 1963, El Chichón in April 1982, and Mount Pinatubo in June 1991 is investigated using 11 global atmospheric reanalysis data sets (JRA-55, JRA-25, MERRA-2, MERRA, ERA-Interim, ERA-40, CFSR, NCEP-NCAR R-1, 20CR version 2c, ERA-20C, and CERA-20C). Multiple linear regression (MLR) is applied to the monthly mean time series of temperature for two periods, 1980–2010 (for 10 reanalyses) and 1958–2001 (for six reanalyses), by considering explanatory factors of seasonal harmonics, linear trends, Quasi-Biennial Oscillation (QBO), solar cycle, tropical sea surface temperature (SST) variations in the Pacific, Indian, and Atlantic Oceans, and Arctic SST variations. Empirical orthogonal function (EOF) analysis is applied to these climatic indices to obtain a set of orthogonal indices to be used for the MLR. The residuals of the MLR are used to define the volcanic signals for the three eruptions separately. First, area-averaged time series of the residuals are investigated and compared with the results from previous studies. Then, the geographical distribution of the response during the peak cooling period after each eruption is investigated. In general, different reanalyses show similar geographical patterns of the response, but with the largest differences in the polar regions. The Pinatubo response shows largest average cooling in the 60°N–60°S region among the three eruptions, with a peak cooling of 0.10–0.15 K. The El Chichón response shows slightly larger cooling in the NH than in the Southern Hemisphere (SH), while the Agung response shows larger cooling in the SH. These hemispheric differences are consistent with the distribution of stratospheric aerosol optical depth after these eruptions; however, the peak cooling after these two eruptions is comparable in magnitude to unexplained cooling events in other periods without volcanic influence. Other methods in which the MLR model is used with different sets of indices are also tested, and it is found that careful treatment of tropical SST variability is necessary to evaluate the surface response to volcanic eruptions in observations and reanalyses.

## 1 Introduction

Explosive volcanic eruptions that inject substantial amounts of $SO_2$ and $H_2S$ directly into the stratosphere have significant impacts on the climate via the radiative forcing effects of stratospheric sulphate aerosol particles that form from these precursor

species (Robock, 2000; see also IPCC, 2013). Such eruptions are particularly influential when they occur in the tropics because the resulting aerosol particles are transported globally through the stratospheric meridional circulation (or the Brewer-Dobson circulation; Butchart, 2014), which redistributes air from low to higher latitudes. The increased concentration of aerosols in the stratosphere causes a net negative radiative forcing at the surface (Robock, 2000), resulting in cold surface temperature

anomalies when averaged globally or over the tropics. The geographical distribution of the surface temperature anomalies is, however, found to be much more complicated. Robock (2000) reviewed observations and theory of wintertime warming over the Northern Hemisphere (NH) continents (or the wave pattern of warm/cold anomalies) that result from changes in the tropospheric and stratospheric circulations after large eruptions. The surface temperature response at the regional scale is thus not only influenced by the direct radiative forcing but also by the dynamical response of the atmospheric circulation. Studies

on the geographical distribution of the surface volcanic response all show complex patterns of cooling and warming (e.g., Kirchner et al., 1999; Yang and Schlesinger, 2001).

After the establishment of the global radiosonde observation network in the late 1950s, there have occurred three major volcanic eruptions that affected the global climate: Mount Agung (8°S, 116°E), Bali, Indonesia in March 1963; El Chichón

(17°N, 93°W), Chiapas, Mexico in April 1982; and Mount Pinatubo (15°N, 120°E), Luzon, Philippines in June 1991. In each of these cases, volcanic aerosols remained in the stratosphere for multiple years and exerted a net negative radiative forcing (see e.g. Chapter 8 and Figure 8.18 of IPCC, 2013, for recent estimates of the radiative forcing due to these eruptions). The surface cooling that resulted following each of these eruptions has been analysed using both observational data (e.g. Angell and Korshover, 1984; Mass and Portman, 1989; Parker et al., 1996; Kelly et al., 1996; Yang and Schlesinger, 2001) and model

outputs in comparison with observational data (e.g. Hansen et al., 1978, 1996; Kirchner et al., 1999; Chapters 9 and 10 of IPCC, 2013; Driscoll et al., 2012; Wunderlich and Mitchell, 2017). Most of these studies have focused on time series of the global mean, the tropical mean, or means in other latitude bands; however, for the Pinatubo case, some studies also analysed the geographical distribution of the response (Hansen et al., 1996; Parker et al., 1996; Kelly et al., 1996; Kirchner et al., 1999; Yang and Schlesinger, 2001). Because El Niño events occurred after all three eruptions (the 1982–1983 El Niño following the

El Chichón eruption being the strongest), all the above studies were aware of the need to remove the El Niño warming component when evaluating the volcanic cooling. However, most studies simply used anomalies from the long-term mean state to address this; only some recent studies (e.g. Yang and Schlesinger, 2001; Wunderlich and Mitchell, 2017) have used SST data to explicitly subtract the El Niño–Southern Oscillation component (ENSO; e.g. Barnston et al., 1997). Moreover, some of the studies analysed composite response for several major eruptions, rather than treating each eruption separately (e.g.

Driscoll et al., 2012; Wunderlich and Mitchell, 2017). Finally, the observational data used in the above studies comprised, in most cases, station data or gridded station data, although global atmospheric reanalysis products were used by Kirchner et al. (1999), Driscoll et al. (2012), and Wunderlich and Mitchell (2017). The current study aims to expand on these previous works by investigating the surface temperature response both in area means and in geographical distributions, treating each of the

three major eruptions separately, explicitly subtracting ENSO and other known forced components, and using global atmospheric reanalysis data.

Global atmospheric reanalysis data sets are produced by assimilating various observational data together with global forecast model outputs using data assimilation techniques, and thus represent 'best estimates' of past atmospheric states. They are among the essential types of gridded data sets used to investigate past weather and climate, including the climatic response to major volcanic eruptions. Currently, several reanalysis data sets have been made available by four reanalysis centres (Table 1; see also Fujiwara et al., 2017). It is well known that different reanalyses give different results for certain diagnostics, giving rise to various intercomparison and validation activities (such as the SPARC Reanalysis Intercomparison Project; Fujiwara et al., 2017). It is therefore of interest to compare how different reanalyses represent the surface temperature responses following the three major volcanic eruptions. In this study, we analyse data from the following 11 reanalyses (see Table 1): ERA-Interim, ERA-40, ERA-20C, and CERA-20C produced by the European Centre for Medium-Range Weather Forecasts; JRA-55 and JRA-25 produced by the Japan Meteorological Agency (in cooperation with the Central Research Institute of Electric Power Industry for JRA-25); MERRA-2 and MERRA produced by the National Aeronautics and Space Administration; and NCEP-NCAR R-1, CFSR, and 20CR version 2c (20CRv2c) produced by the National Oceanic and Atmospheric Administration (NOAA), in cooperation with the National Center for Atmospheric Research for R-1 and with the University of Colorado at Boulder for 20CRv2c.

Fujiwara et al. (2015) investigated the global temperature response in the troposphere and stratosphere to each of the three major volcanic eruptions by applying multiple linear regression (MLR) to zonal and monthly mean temperature fields from 9 global atmospheric reanalyses. By evaluating and subtracting known components of variability in the temperature time series, including ENSO (via the Niño 3.4 index), Quasi-Biennial Oscillation (QBO; Baldwin et al., 2011), and 11-year solar cycle (Tapping, 2013) in addition to seasonal variations and linear trends, they assumed that the residual time series following MLR comprised volcanic signals and random variations. This approach was used to evaluate the response to each of the three major volcanic eruptions separately. Following Fujiwara et al. (2015), we assume that monthly mean 2-metre surface temperature (T2m) variability has components due to the major volcanic eruptions, tropical (and Arctic) sea surface temperature (SST) variations, the QBO, the solar cycle, and seasonal variations and linear trends. We do not consider modes of extratropical atmospheric variability such as the North Atlantic Oscillation (NAO) and the Pacific-North American (PNA) teleconnection pattern as explanatory factors for T2m variability because of the possibility that the midlatitude response to volcanic forcing projects on these modes through shifts in the jet stream (Dallasanta et al., 2019).

While conducting this study on the surface temperature response to the major volcanic eruptions, we found that we needed to adopt more careful approaches for treating tropical SST variations relative to that used by Fujiwara et al. (2015). First, El Niño events with different spatial patterns occurred in either year one or year two after all three major eruptions. The 1963–1964 El

Niño event showed a peak SST warming in December 1963 located roughly in the Niño 3.4 region (see Table 2 for definitions of the Niño regions). Meanwhile, the 1982–1983 El Niño event showed a peak warming in January 1983 covering both the Niño 3 and Niño 1+2 regions, and the 1991–1992 El Niño event showed a peak warming in January 1992 in the Niño 3.4 region. Thus, it is not appropriate to use a single Niño index to account for all ENSO signals in the MLR analysis. Second,

there are a number of other known large-scale modes of variability in tropical SST, such as El Niño Modoki in the central Pacific (Ashok et al., 2007), the Indian Ocean basin mode (Zheng et al., 2011) and Indian Ocean dipole mode (Saji et al., 1999) in the tropical Indian Ocean (see also Guo et al., 2017), and variations in the Atlantic cold tongue (a.k.a. Atlantic Niño) or the northern tropical Atlantic (Richter et al., 2013; see also Xie and Carton, 2004). Tracking and subtracting the components related to these SST variations in addition to ENSO may better isolate the volcanic signals. In addition, recent studies have

revealed a wintertime teleconnection rooted in the Arctic Ocean that influences East Asia and North America (Mori et al., 2014, 2019; Kug et al., 2015). In test calculations using two Arctic Ocean SST indices (not only in winter, but throughout the year) suggested by Kug et al. (2015), we find some small improvements in reconciling the volcanic signals. We therefore consider the nine tropical and two Arctic SST indices listed in Table 2. Together with two QBO indices and a solar cycle index, we apply empirical orthogonal function (EOF) analysis (e.g., Dommenget and Latif, 2002) to these 14 climatic indices to

obtain a set of 14 orthogonal indices that we then use in the MLR. EOF analysis has been previously used in a similar way to obtain two orthogonal QBO indices (e.g., Yamashita et al., 2018). As in Fujiwara et al. (2015), residuals from the MLR are used to define the volcanic signals for the three eruptions separately. Alternative methods using different sets of climatic indices have also been tested; these sensitivity studies are discussed in Appendices A and B.

We note that our approach to isolate the surface temperature anomalies associated with volcanic eruptions from other external forcings is imperfect. One important limitation is that a fraction of ENSO-related variability may emerge from the volcanic response through forced changes in atmospheric circulation (e.g., Wang et al., 2018, and references therein). Our method implicitly assumes that the zonally-symmetric volcanic aerosol forcing does not project substantially onto strongly asymmetric modes of variability like ENSO. However, the impacts of volcanic eruptions on ENSO-related variability and other modes of

coupled atmosphere–ocean variability are not well characterized and thus some uncertainties related to this influence remain in our analysis. The temperature anomalies following volcanic eruptions as reported below should be regarded as the component of the volcanic forcing that is not mediated by coupled modes of atmosphere–ocean variability.

The remainder of this paper is organized as follows. In Section 2, we describe the 2-metre temperature data products from the

11 reanalysis data sets. In Section 3, we introduce the MLR analysis method as applied in this paper. We present and discuss the results of the analysis in Section 4, with concluding remarks in Section 5. In Appendices A and B, we provide brief summaries and comparisons of selected results using alternative methods, which serve to validate the primary choice of method.

## 2 Data Description

Table 1 lists the 11 global atmospheric reanalysis data sets that are analysed in this study, including reference information. Summary descriptions of the forecast model, assimilation scheme, and assimilated observations for each of these reanalyses have been provided by Fujiwara et al. (2017) and Zhou et al. (2018). The reanalyses ERA-Interim, ERA-40, JRA-55, JRA-25,

5   MERRA-2, MERRA, CFSR, and R-1 are "full-input" reanalyses that assimilate surface and upper-air conventional and satellite observational data, with satellite data from around the early 1970s but mainly from 1979 onward (for satellite data, R-1 assimilates only retrievals, while others assimilate both radiances and retrievals). By contrast, the reanalyses ERA-20C, CERA-20C, and 20CRv2c are "surface-input" reanalyses that assimilate surface data only (pressure for all; marine winds for ERA-20C and CERA-20C). Excluding upper-air observations enables the latter three reanalyses to cover much longer periods.

Station observations of 2-metre temperatures are assimilated only in ERA-Interim, ERA-40, JRA-55, and JRA-25. Analysis of surface air variables in these reanalysis systems is conducted via univariate two-dimensional optimal interpolation analysis steps that are separate from the standard three-dimensional or four-dimensional variational analysis cycles. Monthly means for these reanalyses represent averages of four-times-daily products. None of the other reanalyses considered in this paper

assimilate surface-air station observations. For R-1, CFSR, 20CRv2c, ERA-20C, and CERA-20C, the 2-metre temperature products are derived primarily from the forecast model, although are still affected by the assimilation of other observations (i.e. surface pressure and, in the case of full-input reanalyses, upper-air measurements). For R-1, monthly means of four-times-daily data (in this case consisting of 6-hour forecasts) are provided for 2-metre temperature. For CFSR, we use the "flxf06" monthly mean products for consistency with R-1 data, as these files also contain 6-hour forecast outputs. For 20CRv2c,

monthly means of eight times daily data (the 3-hour and 6-hour forecasts from 00/06/12/18 UTC) are provided (G. Compo, private communication, 2018). For ERA-20C (CERA-20C), monthly means of four (eight) times daily data are provided (H. Hersbach, private communication, 2018). Note that for CERA-20C, monthly means are provided for each of the 10 ensemble members; in this study, we calculate and analyse averages of the 10 members. For MERRA-2 and MERRA, monthly mean 2-metre temperatures in the "tavgM_2d_slv_Nx" data files are analysed. These products are monthly means of the "assimilation"

(or "ASM") product (at all model time steps), which represent the analysis state after the incremental analysis update (IAU) procedure is applied (see Rienecker et al., 2011, their Figure 1). In the IAU, the analysis correction is applied to the forecast model gradually, and thus the ASM product is distinct both from the initial forecast and the 6-hourly analysis ("ANA") product. The 2-metre temperature data in MERRA and MERRA-2 are affected by the observational analysis (but not by 2-metre temperature observations, which are not assimilated) through the lowest model level, interpolated to 2-m height using a Monin-

Obukov method (Molod et al., 2015).

Different reanalyses use different SST and sea ice data sets as lower boundary conditions (see references in Table 1; see also Table 4 of Fujiwara et al., 2017). The treatment of aerosols also differs amongst the reanalyses (see also Section 3.2 of Fujiwara

et al., 2017). The radiative effects of volcanic aerosols are considered in the forecast models of CFSR, 20CRv2c, ERA-20C, CERA-20C, and MERRA-2 only. For other reanalyses, any volcanic response in meteorological fields is entirely due to the influences of assimilated observations. Fortunately, all the reanalyses show reasonable volcanic signals in the atmosphere (Fujiwara et al., 2015) despite larger departures between the observations and background forecasts (without accounting for

volcanic aerosols) during periods with strong volcanic influence. The impacts of different volcanic aerosol treatments on radiative fluxes in various reanalyses has been examined by Bosilovich et al. (2015; their Figures 4-1 and 4-3). Finally, we note that MERRA-2 uses observation-corrected precipitation as an input to the land model (Reichle et al., 2017), which helps to constrain the surface latent heat flux and surface energy storage as these relate to temperature. Although this generally impacts positively on the 2-metre temperature product, in cases where few precipitation observations are available (particularly

over Africa) there may be large uncertainties in the forcing (M. Bosilovich, private communication, 2018).

Basic intercomparisons of 2-metre temperatures across reanalyses have been provided by Simmons et al. (2017), Draper et al. (2018), and Zhou et al. (2018), as well as in the references listed in Table 1.

**3 Method**

An MLR analysis is used to remove non-volcanic signals from the time series. Following Fujiwara et al. (2015), two MLR analysis periods are defined based on data availability as shown in Table 1. The first analysis period covers 1980–2010 (31 years) and includes 10 reanalyses (all except ERA-40). The second analysis period covers 1958–2001 (44 years) and includes six reanalyses (ERA-40, ERA-20C, CERA-20C, JRA-55, R-1, and 20CRv2c). These periods are chosen because MERRA-2

started in January 1980, CFSR ended in December 2010 (transitioning to the updated CFSv2 system in January 2011), JRA-55 started in January 1958, and ERA-40 ended in August 2002. Also, MERRA-2 stops assimilating volcanic eruption information in 2010 (Randles et al., 2017). The first analysis period covers the eruptions of El Chichón in 1982 and Mount Pinatubo in 1991, while the latter covers these two eruptions as well as the eruption of Mount Agung in 1963.

The MLR model that we use in this study is

$$Y(t) = a_0 + \sum_{l=1}^{N} a_l \, x_l(t) + R(t)$$

where $Y(t)$ is the monthly mean time series of surface temperature at a particular longitude–latitude grid point, $a_l$ is the least squares solution of a fitting parameter for the index time series $x_l(t)$, and $N$ is the total number of indices (i.e., potential explanatory factors). $R(t)$ is the residual of the model, which is assumed to consist of volcanic signals and random variations

as in Fujiwara et al. (2015). For indices representing the seasonal cycle, we use six seasonal harmonics of the form, $a_1 \, \sin \omega t + a_2 \, \cos \omega t + a_3 \sin 2\omega t + a_4 \cos 2\omega t + a_5 \sin 3\omega t + a_6 \cos 3\omega t$ , with $\omega = 2\pi / (12 \text{ months})$. We further consider the

linear trend, along with 14 "EOF" indices (explained below). The latter are based on the 11 SST indices listed in Table 2 (as discussed in Section 1), two QBO indices, and a solar cycle index. These indices are introduced in the following paragraph. In practice, the six seasonal harmonics and a constant are considered for each of the 15 indices, resulting in seven combinations for each of the core indices as in Fujiwara et al. (2015). Thus, $N$ is $6 + 15 \times 7 = 111$.

The NOAA Extended Reconstructed Sea Surface Temperature version 5 (ERSSTv5) data set (monthly, $2° \times 2°$ grid) is used to construct the 11 SST indices listed in Table 2. This is because the NOAA Climate Prediction Center (CPC) uses this data set to construct their monthly Niño indices. The 30-year base period to obtain SST anomalies is chosen as 1981–2010 for both the 1958–2001 and 1980–2010 MLR analyses. A cosine-of-latitude weighting is considered when calculating regional averages. We have confirmed that the Niño indices calculated from ERSSTv5 are essentially identical to those obtained from the NOAA CPC. For the two QBO indices, we use 20 hPa and 50 hPa monthly-mean zonal wind data from four equatorial radiosonde stations, including Canton Island (2.8°S, 171.7°W; January 1953–August 1967), Gan, Maldives (0.7°S, 73.2°E; September 1967–December 1975), and Singapore (1.3°N, 104°E; January 1976–present), as compiled and published by the Freie Universität Berlin. For the solar cycle index, we use monthly averages of solar flux density data (the "absolute" flux data) at 2.8 GHz (or at 10.7 cm) obtained by radio telescopes located near Ottawa, Canada (February 1947–May 1991) and Penticton, Canada (June 1991–present) (Tapping, 2013).

The explanatory factors (or indices) used in the MLR analysis should ideally be mutually orthogonal. Thus, we detrend the 11 SST indices, the two QBO indices, and the solar cycle index for each of the two MLR analysis periods and then apply EOF analysis to the detrended time series to obtain two sets of 14 orthogonal indices. The MLR analysis is then conducted separately for each reanalysis data set and period. The volcanic signals are then defined as the three-month averages (or three-month running averages in time series figures) of the residual time series $R(t)$. Note that, for simplicity, we do not adopt a base period for the volcanic signals as was done by Fujiwara et al. (2015). In Appendix A, we validate this method against results obtained from different approaches to prescribing the indices used in the MLR analysis.

## 4 Results and Discussion

### 4.1 The 1980–2010 analysis

Figure 1 shows time series of the residuals from the 1980–2010 MLR analysis, which include volcanic signals and random variations, averaged over the 60°N–60°S and 60°N–Equator latitude bands for each of the 10 reanalyses. Figure 1 also includes time series of stratospheric aerosol optical depth (AOD), which show that the stratospheric aerosol layer following the Mount Pinatubo eruption extended to both hemispheres (with slightly larger amounts in the NH), while that following the

El Chichón eruption was more confined to the NH (see also Figure 9c of Fujiwara et al., 2015 for the latitudinal distribution of the stratospheric AOD). In both cases, the peak cold anomaly occurred about one and a half years after the corresponding eruption. For the Mount Pinatubo eruption, the magnitude of this anomaly reached 0.10–0.15 K around September–November (SON) 1992 (or December 1992–February 1993) in 60°N–60°S averages, although this signal was somewhat stronger and had a slightly longer duration in the 60°N–Equator averages. For the El Chichón eruption, the anomaly was smaller and less distinctive relative to other unexplained random variations, although the cooling reached ~0.10 K around June–August (JJA) 1983 in the 60°N–Equator averages. We observe both warm and cold anomalies during periods without volcanic influence, which are nonetheless unexplained by the indices considered in our MLR analysis. Inclusion of the two Arctic-Ocean SST indices reduced the amplitudes of some of the anomalies (as noted in the Introduction), but anomalies of comparable magnitude still remain.

The peak cooling amplitudes obtained here may be smaller than those reported by previous studies that used simpler methods to evaluate the volcanic signals. For example, Figure 9.8 of IPCC (2013) shows long-term time series of observed (as well as simulated) global mean surface air temperature using three gridded data sets (not reanalyses) as anomalies from the corresponding 1961–1990 time means. Such anomaly time series roughly indicate warming trends, temporary cooling events after major volcanic eruptions, and other variations with time scales and magnitudes similar to those of volcanic signals. The peak values of volcanic cooling as shown in Figure 9.8 of IPCC (2013) approach ~0.2 K for all three major eruptions. Furthermore, Figure 10.6 of IPCC (2013) (see also Imbers et al., 2013) shows results from four different regression analyses onto a volcanic aerosol index, an ENSO index, and other selected indices using gridded global mean temperature anomalies relative to the 1980–2000 time mean. This presentation indicates that the peak cold anomalies differed among the three eruptions depending on the corresponding values of the volcanic aerosol index (see Figure 1c): ~0.15–0.2 K for the Pinatubo case and ~0.1–0.15 K for the other two cases. Earlier studies (e.g., Parker et al., 1996; Hansen et al., 1996) claimed even greater global surface cooling of ~0.5 K for the Pinatubo case. The smaller values in our Figure 1 may emerge mainly from differences in methodology. We have endeavoured to subtract all known forced components other than the major volcanic eruptions, resulting in generally smaller variance in the residual (including the volcanic signals). Furthermore, taking global means may increase uncertainties relative to the latitude bands shown in Figure 1 because uncertainties in temperature data are largest in the high-latitude and polar regions, as shown below.

Figure 2 shows geographical distributions of the 2-metre temperature anomalies averaged for SON 1992 (following the Mount Pinatubo eruption in June 1991) for each of the 10 reanalyses. Although previous studies have discussed "winter" and "summer" anomalies separately, we focus primarily on the 3-month period when the peak cooling occurred for each eruption (i.e. SON 1992 for the Pinatubo case; however, see Figure 3 for the first NH winter response and the second-year NH summer anomalies following the Pinatubo eruption based on two reanalyses). In Figure 2 (and in all other figures showing geographical distributions), coloured shading marks regions with positive or negative values of the 3-month mean residual

$R(t)$ with absolute magnitudes that exceed one standard deviation (SD) of $R(t)$ at that location. All reanalyses produce smaller SD values over the tropical and mid-latitude oceans and larger SD values over the continents. The largest SD values among continental regions are in the NH high latitudes and over Antarctica, while the largest SD values for maritime regions are over the Arctic Ocean and the Southern Ocean. The JRA-55 result (top left) shows the following characteristics: (1) the equatorial 10°N–10°S region shows weak cooling over most of the oceans and warming over the eastern part of the Indonesian maritime continent; (2) the 15°N–70°N region shows cooling on average, comprising stronger cooling over the midlatitude Atlantic, northern Africa, the northern Atlantic Ocean into western Europe, and East Asia but warming over the eastern Pacific; (3) the 10°S–60°S region shows weak cooling on average, comprising stronger cooling over South America and the subtropical eastern Indian Ocean to Australia but warming over other regions; (4) the 70°N–80°N Arctic region shows warming except in the Eurasian sector; (5) the Southern Ocean shows a wave one pattern with cooling centred around the Greenwich meridian; and (6) Antarctica shows cooling in general but with the strongest cooling centred around 90°W–0°. All other reanalyses in Figure 2 show very similar cooling/warming patterns, though with some differences in magnitude (especially at higher latitudes).

Hansen et al. (1992, their colour Figure 4) showed the SON 1992 anomalies "predicted" using a global climate model with two scenarios (representing starting the model run on two different dates). Although the geographical pattern of the anomalies cannot be directly compared, we find generally similar horizontal scales in the cooling and warming regions globally. Hansen et al. (1996), Parker et al. (1996), Kelly et al. (1996), Kirchner et al. (1999), and Yang and Schlesinger (2001) showed geographical distributions of the "observed" anomalies averaged over the first NH winter (December 1991 to February 1992, D91–JF92). Three studies out of the five (i.e., except Parker et al., 1996, and Kelly et al., 1996) also showed the anomalies for the second-year NH summer (June to August (JJA) 1992). Yang and Schlesinger (2001) explicitly removed ENSO signals, while the other four studies simply showed anomalies relative to long-term means. In Figure 3, we show the anomalies for these two periods using JRA-55 and R-1 (the latter being equivalent to the "NCEP reanalysis" as used by Kirchner et al., 1996). Other reanalyses show similar results. For the D91–JF92 response, we see a widespread region of strong warming region over Eurasia, which appears in results based on both JRA-55 and R-1 (and all other reanalyses examined in this study; not shown), as well as in the five previous studies. We note also a prominent cooling signal in the Arctic (over Barents-Kara Seas and Greenland Sea), which together with the warm Eurasian signal form the opposite of the "warm Arctic–cold continent" pattern observed in recent surface temperature trends (Mori et al., 2014). Interannual variability in sea ice concentration is also associated with a similar surface temperature response (Chen et al., 2016) and may not have been completely removed from the residual by the MLR (which uses SST-based indices but not sea-ice-based indices in the Arctic sector). Over North America, on the other hand, our results (in JRA-55, R-1, and all other reanalyses) show mainly cooling (with warming around 30°N) during D91–JF92. This result contrasts with previous studies, which showed warming in this region. Yang and Schlesinger (2001, their Figure 6) have removed ENSO components from surface temperature over North America but still report warming in this region. Our consideration of the whole spectrum of tropical Pacific variability may be responsible for this difference

relative to earlier studies. Further discussion on this topic is provided in Appendix B. For the JJA 1992 anomalies, we see a global cooling in general, but with four regions showing warm anomalies instead: the NH mid-latitude eastern Pacific, western Europe, the SH mid-latitude eastern Pacific, and most of the ~70°S latitude band (except near the Greenwich meridian). The studies by Hansen et al. (1996), Kirchner et al. (1999; showing the region ~20°S to the North Pole), and Yang and Schlesinger

(2001; showing the NH continents) showed broadly similar patterns.

Figure 4 shows the anomalies averaged for JJA 1983, following the El Chichón eruption in April 1982. The JRA-55 result (top left) shows the following characteristics: (1) the equatorial 10°N–10°S region shows a mixture of weak cooling and warming signals, with relatively distinct cooling over western equatorial Africa; (2) the 15°N–90°N region shows cooling on average,

with patchy cooling over the eastern Pacific and Greenland, among other regions; (3) the 20°S–50°S region again shows a mixture of cooling and warming signals, with cooling over the eastern edge of the Pacific into South America and the western edge of the Pacific but warming over the central Pacific and western Australia; and (4) the 90°W–0° sector of the Southern Ocean shows strong warming, while the 90°E–180°–90°W sector of Antarctica and the Southern Ocean shows strong cooling. All other reanalyses in Figure 4 show broadly similar cooling and warming patterns, though with some differences in

magnitude in the polar regions (especially over Antarctica).

To the authors' knowledge, no previous study has shown the geographical distribution of the response to the El Chichón eruption. Note that a very strong El Niño event during 1982–1983 masked the cooling response to the eruption (e.g., Angell and Korshover, 1984). See also Appendix A for detailed discussion on the need for careful treatment of ENSO signals.

## 4.2 The 1958–2001 analysis

Figure 5 shows time series of the residuals from the 1958–2001 MLR analysis, averaged over the 60°N–60°S and Equator–60°S (i.e., the SH) domains for each of the six reanalyses. AOD time series (Figure 5c; see also Figure 9c of Fujiwara et al., 2015) indicate that stratospheric aerosols following the Mount Agung eruption were located primarily in the SH (see also

Fujiwara et al., 2015, who showed lower stratospheric warming from the mid-latitude SH to the tropics following the Agung eruption). For the Mount Pinatubo eruption, the 60°N–60°S average shows a peak cooling with similar timing and similar magnitude to that from the 1980–2010 MLR analysis (Figure 1a). The Equator–60°S average also shows a clear cooling signal but with somewhat different magnitudes in the different reanalyses. The cooling signal following the El Chichón eruption is not very distinct relative to other unexplained warming and cooling signals, although the 60°N–60°S average does show a

cooling of similar magnitude to that implied by the 1980–2010 MLR analysis (Figure 1a). The Mount Agung eruption was followed by a cooling signal of ~0.15 K, which is especially apparent in the Equator–60°S average. However, the timing of the peak cooling is rather ambiguous, owing in part to larger differences among the reanalyses and in part to the possible existence of a slightly later second peak.

A handful of previous studies have discussed or shown the global-mean anomalies associated with the Mount Agung eruption (e.g., Hansen et al., 1978; Angell and Korshover, 1984; Mass and Portman, 1989; Imbers et al., 2013; IPCC, 2013). As with the Pinatubo and El Chichón cases discussed in the previous section, older studies tend to report larger cooling anomalies. As above, our results are based on a more sophisticated method for isolating the volcanic signal and avoid the use of global means (which are subject to larger uncertainties).

Figure 5 also shows a transient cooling event in 1976 that is remarkably consistent among the reanalyses. This cooling may be related to the eruption of Mount Fuego (14°N, 91°W), Guatemala, in October–December 1974 (Smithsonian Institution National Museum of Natural History Global Volcanism Program, http://www.volcano.si.edu/, last accessed March 2015). The stratospheric aerosol loading peaked in 1975, though the peak AOD was much smaller than that following the three major eruptions discussed in this work. However, Mass and Portman (1989) expressed doubt that this eruption had any notable influence on the surface temperature. Moreover, Figure 7 of Fujiwara et al. (2015) found no relevant signal in the upper troposphere and lower stratosphere during this period, except in ERA-40, which shows very different (and most probably unrealistic) signals there. Therefore, the cause of the transient cooling event in 1976 needs further investigation.

Figure 6 shows the SON 1992 anomalies associated with the Mount Pinatubo eruption (as in Figure 2) from the 1958–2001 MLR analysis for the six reanalyses. The JRA-55 result (top left) shows the same general characteristics as described for Figure 2 in the previous section, with the exception of substantially weaker warming signals in the Arctic region. Comparison of zonal mean results for JRA-55 between Figures 2 and 6 shows that the NH cooling signal is larger in Figure 6. All other reanalyses in Figure 6 show similar geographical patterns to JRA-55, but with some substantial discrepancies in the polar regions.

Figure 7 shows the JJA 1983 anomalies associated with the El Chichón eruption (as in Figure 4) from the 1958–2001 MLR analysis. The JRA-55 result (top left) again shows the same general characteristics in the equatorial region and in the SH as described for Figure 4; however, the cooling regions at NH middle and high latitudes are substantially wider and larger in amplitude when compared with the results in Figure 4. All other reanalyses in Figure 7 show similar characteristics to JRA-55.

The 1958–2001 analysis provides us not only the Mount Agung anomalies but also an opportunity to test the robustness of the results following the eruptions of El Chichón and Mount Pinatubo. Both the similarities and the differences between Figures 2 and 6 and between Figures 4 and 7 for any given reanalysis data set (JRA-55, R-1, 20CRv2c, ERA-20C, or CERA-20C) highlight the extent to which the results depend on the choice of the MLR analysis period (i.e. 1980–2010 versus 1958–2001). The geographical patterns of cooling and warming signals described in the previous section are generally robust between the

two analysis periods, though we find that the NH part of the JJA 1983 anomalies associated with the El Chichón eruption is relatively sensitive to the choice of analysis period with respect to the magnitude of the cooling.

The cooling anomalies associated with the Mount Agung eruption in March 1963 averaged over the Equator–60°S region (Figure 5b) differs more among the reanalyses than the anomalies associated with the other two major eruptions, with some reanalyses showing double cooling peaks during the ~1.5 years following the eruption. The geographical distribution of the 3-month running mean anomalies following the eruption shows two subtropical-to-midlatitude cooling regions centred at ~30°S, one in the Atlantic and the other ranging from Africa to the western Indian Ocean. Both anomalies are quite persistent through the ~1.5 years immediately following the eruption. Figure 8 shows the spatial pattern of the anomalies averaged over June to August 1964. The JRA-55 result (top left) shows the following characteristics: (1) signals in the tropical region are relatively weak but with numerous small warming regions; (2) the NH subtropical and mid-latitude regions show several relatively strong centres of both signs; (3) the SH subtropical and mid-latitude regions show two distinct cold anomalies, one in the Atlantic to South America and the other extending eastward from Africa into the western Indian Ocean, while Australia and parts of the Pacific show scattered warm anomalies; (4) the signals at NH high latitudes are varied and weak; and (5) the SH high latitudes show strong warming outside of some sectors of the Southern Ocean. The other reanalyses in Figure 8 show broadly similar characteristics except in the SH polar region where differences among the reanalyses are quite large. These large differences are probably due to generally low availability of observational data in this region.

To the authors' knowledge, no previous study has shown the geographical distribution of the anomalies associated with the Agung eruption. The major cooling regions in Figure 8 are located over the oceans in the subtropical-to-midlatitude SH where sparse ship measurements would be the main data source, yet the six reanalyses show very similar characteristics in this regard. The radiative and dynamical processes that are responsible for producing this spatial pattern of cooling in the SH (i.e. ocean cooling or planetary-scale wave patterns) should be explored in future work.

**5 Conclusions**

In this study, we have evaluated the surface temperature response to each of the three major eruptions that occurred during the latter half of the 20th century separately. We have used 11 global atmospheric reanalysis data sets for the purpose of intercomparing different reanalyses and assessing uncertainties related to inter-dataset differences in representations of atmospheric processes. We have used an MLR analysis technique to estimate and eliminate all known forced components of surface temperature variability (i.e. those that are not regulated by dynamics intrinsic to the lower-tropospheric circulation), and have adopted an EOF analysis technique to convert the relevant climatic indices to an orthogonal set of indices. The residual time series is assumed to comprise the effects of volcanic signals and internal variability in the lower-tropospheric

circulation on 2-metre temperature. We suggest that this method, which is here used for the first time in studying the climatic response to volcanic eruptions, is a viable approach for isolating volcanic signals from contamination by other factors that modulate surface temperature variability, such as ENSO and other coupled modes of atmosphere-ocean variability. We provide a more detailed discussion of the methodology and its performance in Appendix A. However, we note that the residual time series still contains several cooling and warming signals similar in magnitude to the volcanic signals during periods without extensive volcanic influence. These signals may arise from natural, unforced variability, but could also include some components of forced variability that are as yet unrepresented in our MLR analysis. As discussed in the Introduction, our method has limitations. In particular, our method neglects possible responses of ENSO or other major modes of internal variability to the eruptions, which may cause our residual term to underestimate the scale of the response.

We have investigated the geographical distribution of the surface air temperature anomalies following the major volcanic eruptions, as well as area-averaged time series and zonal means. To our knowledge, this work is the first time that the geographical distributions of anomalies associated with the El Chichón or Mount Agung eruptions have been extensively investigated.

Figure 9 shows inter-reanalysis differences in the geographical distribution of the anomalies for four cases. Figure 9 shows that differences among different reanalyses are generally small outside of the polar regions, with larger differences over the continents than over the oceans. Indeed, differences among different analysis methods (i.e. different MLR analysis periods or different sets of climatic indices) are larger than those among the reanalyses. The results for the Mount Agung eruption in 1963 (before the introduction of satellite microwave and infrared sounders in the 1970s) show larger differences among the reanalyses than do the results for the two more recent eruptions in 1982 and 1991. Large inter-reanalysis differences in the polar regions persist even for the Mount Pinatubo case, likely due to the continuing paucity of observations there. Indeed, the largest inter-reanalysis differences in the results for all three cases are located at high latitudes, and especially in the SH.

In comparison with previous studies, our zonal-mean results tend to imply smaller cooling magnitudes following the major volcanic eruptions. We have more thoroughly considered potential confounding factors outside of the volcanic eruptions themselves. The anomalies may be underestimated by our method if the volcanic eruptions analyzed in this study directly influenced ENSO variability (e.g., Wang et al., 2018, and references therein). However, at the very least, we argue that a global cooling of ~0.5 K as claimed by studies from the 1990s (e.g. Hansen et al., 1992; Parker et al., 1996) and referred to in more recent discussions on geoengineering (e.g. Crutzen, 2006) overestimates the actual response to the Pinatubo eruption. This contention is also supported by Figures 9.8 and 10.6 of IPCC (2013) (see also discussion in the second paragraph of section 4.1 above). More appropriate values may be closer to our results, i.e. 0.10–0.15 K for the 60°N–60°S mean, although including polar regions would amplify the uncertainty in this estimate.

The geographical distributions of 2-metre temperature anomalies following the major volcanic eruptions show complicated patterns but are nonetheless quite similar to the results of previous studies that investigated surface temperature anomalies associated with the Mount Pinatubo case. Anomalies over North America are a notable exception. Whereas previous studies showed warming over most of North America, we find extensive cooling. This difference is likely due to our inclusion of the whole spectrum of tropical Pacific variability, especially the Niño 3 and Niño 3.4 regions, in the MLR analysis (see Appendix B for details). The warming anomalies over the NH continents in the boreal winters following the Mount Pinatubo eruption have attracted much attention (e.g. Robock, 2000; Wunderlich and Mitchell, 2017), but may require a closer look in future work. The geographical distributions of surface air temperature anomalies following the El Chichón and Mount Agung eruptions are also very interesting. Mid-latitude planetary-scale wave patterns corresponding to a variety of zonal wave numbers seem to be the common characteristics, suggesting that atmospheric dynamics play an important role in the response.

Finally, in the context of our results, we comment briefly on solar radiation management (SRM), one of the more commonly proposed categories of climate engineering or geoengineering (e.g., Crutzen, 2006; Chapter 7 of IPCC, 2013). As noted above, an estimated global cooling of ~0.5 K for the Pinatubo case may be up to ~5 times too large. Furthermore, uncertainties in the response are greater in the polar regions than at lower latitudes, and very careful data analysis procedures are needed when evaluating and subtracting non-volcanic (or non-SRM) components. Thus, evaluating the effects of SRM, if implemented, would not be an easy task in the real climate system.

*Code and data availability*

The codes used in this paper can be obtained from the authors upon request. Codes for the MLR and related calculations and for plotting were written in Fortran by MF, and depend on the libraries NXPACK (for handling netCDF files), Linear Algebra PACKage (LAPACK; for matrix operations), and GFD-DENNOU (for plotting). Codes for the EOF and SVD analyses were written in MATLAB by PM. The data sets used in this paper can be obtained from the online archives listed in Tables 3 and 4.

**Appendix A: Comparisons with results from other methods**

To investigate the sensitivity of our results to the choice of climatic indices, we compare the primary method used in the main body of the paper with other plausible approaches. In particular, we apply the MLR model with different sets of indices for the period 1980–2010. All sensitivity tests use the JRA-55 data set for ease of comparison.

The first alternate method (designated the "SVD method" in the following) applies a singular value decomposition (SVD) analysis (e.g., Wallace et al., 1992) to JRA-55 2-metre temperature (T2m) data and ERSST v5 SST data to obtain the first 10 cross-covariance components. Both inputs are given as anomalies from the 1981–2010 climatology. We note that Yang and Schlesinger (2001) used SVD analysis on some limited regions (i.e. surface temperature data over some continents and SST data in the tropical Pacific), whereas we apply the SVD analysis globally (though ocean-only for the SST data, of course). This approach produces 10 time series (of the SST coefficients) that describe the major co-variations of T2m and SST (83.67% of the total variability). These time series are then used to replace the set of specified SST indices used in the primary method as described in the main body of the paper. All further procedures for the SVD method are the same as for the primary analysis. An EOF analysis is constructed using the 10 cross-covariance time series, the two QBO indices, and the solar cycle index, obtaining 13 orthogonal indices. These 13 orthogonal indices are then used together with the linear trend and seasonal harmonics in the MLR analysis as described by Equation (1), with the total number of indices $N$ as $6 + (1+13) \times 7 = 104$. We have also tested the SVD method using the first 20 components (93.14% of the total variability) rather than only the first 10. We briefly discuss the results of both approaches below.

The second alternate method (designated the "single-Niño method" in the following) simply uses a single Niño index (any of Niño 1+2, Niño 4, or Niño 3.4) to describe the SST variability. MLR analysis is then conducted using this single Niño index together with the linear trend, the two QBO indices, the solar cycle index, and the seasonal cycle. The total number of indices $N$ is then $6 + (1+4) \times 7 = 41$. EOF analysis is not used. The results from this single-Niño method highlight the fact that different El Niño events have different characteristic spatial patterns and thus clearly illustrate the need to use a more sophisticated method, such as the primary method used in the main body of the paper or the SVD method described in the preceding paragraph.

Figure A1 compares the SON 1992 anomalies following the Mount Pinatubo eruption based on JRA-55 using five different methods: the one used in the main body of the paper (Figure A1a), the SVD method (Figure A1b), and the three different single-Niño methods (Figure A1c-e). The primary method and the SVD method show generally similar characteristics in both the geographical pattern and the zonal-mean anomalies, although the amplitudes are typically smaller when the SVD method is used. The three single-Niño methods produce mutually similar results globally, but with small differences in the equatorial central Pacific. The implied anomaly in the equatorial central Pacific is cooling in the single-Niño approach when the Niño 4 index is used (qualitatively consistent with the results of the first two methods), while signals are near zero when the single-Niño approach is used with either Niño 1+2 or Niño3.4. The SON 1992 period corresponds to a neutral phase after an El Niño event that reached its peak amplitude in the boreal winter of 1991–1992. The three single-Niño methods also give much stronger cooling signals globally (except for in SH high latitudes) relative to the first two methods. We interpret this by noting that the results from the single-Niño methods contain components due to SST variability in the tropical Atlantic and Indian

Oceans, as well as in the Arctic Ocean. The apparent stronger cooling signals may thus not be attributed solely to the influence of the stratospheric volcanic aerosol layer. We therefore need to consider these additional non-ENSO components of SST variability when evaluating the volcanic signals in 2-metre temperature data.

Figure A2 compares the JJA 1983 anomalies associated with the El Chichón eruption in JRA-55 data using the five different methods. The SVD method again shows generally similar characteristics in both the geographical pattern and the zonal-mean anomalies to the primary method; however, in this case the anomalies are typically larger rather than smaller. The three single-Niño methods again give mutually similar results except for in the tropics, where two widespread regions of very strong warming are observed in the tropical eastern Pacific and in the tropical Indian Ocean for single-Niño methods using the Niño
4 and Niño 3.4 indices. This period coincided with a very strong El Niño event with maximum warming in the Niño 1+2 region. This event is effectively removed by the single-Niño method using the Niño 1+2 index (as is, somehow, the tropical Indian Ocean warming), but is retained by the other two single-Niño methods. Given these discrepancies, we conclude that we should consider all four Niño indices when attempting to remove ENSO-related variability from 2-metre temperature data.

Figure A3 shows the residual time series from the SVD method in comparison with that from the primary method based on data from JRA-55. The cooling signals following the two volcanic eruptions agree well between the two methods. The unexplained cooling and warming peaks during periods without volcanic influence also generally agree between the two methods, although the magnitude is sometimes larger in one than in the other. This suggests that using the first 10 2mT–SST cross-covariance components from SVD analysis is roughly equivalent to using the nine tropical and two Arctic SST indices.
We have also tested results based on the first 20 (instead of 10) cross-covariance components using the SVD method. In this case both the unexplained peaks and the volcanic cooling signals are reduced in magnitude by approximately half. The latter might imply that the 11th to 20th components include volcanic signals common to both T2m and SST. This indicates a limitation of the SVD method for isolating the volcanic signal, as it does not distinguish atmosphere-driven SST variability from ocean-driven T2m variability. The 11 SST indices used in our primary method, on the other hand, were carefully chosen
to represent modes of variability that are strongly influenced by the ocean. This method is also imperfect as these modes often result from coupled interactions between the atmosphere and ocean (Deser et al., 2010) and may therefore respond to volcanic forcing. However, it may be reasonable to assume that these modes, which depend on processes that are highly asymmetrical in the zonal direction, respond weakly to the zonally-symmetric volcanic forcing.

In summary, the differences among the different methods are generally much greater than the differences among different reanalysis data sets shown in the main body of this paper. The single-Niño method cannot be used, and is especially problematic for studies that analyse more than one volcanic eruption via MLR analysis. This is because different El Niño events exhibit different patterns of warming in the tropical eastern Pacific, and therefore cannot be adequately described by a single Niño index. Known modes of interannual SST variability in the tropical Atlantic and Indian Oceans can also force variability in the

surface air and lower troposphere. Moreover, as discussed in the Introduction, conditions over East Asia and North America are known to respond to forcings that emerge from the Arctic Ocean. EOF analysis is a very useful tool for obtaining an orthogonal set of indices from partially-correlated indices (such as the four Niño indices) as well as other indices that may or may not be mutually independent (e.g. SST variability in regions other than the tropical Pacific, the QBO, and the solar cycle).

The SVD method is also a viable candidate for dealing with the limitations of the simple single-Niño approach. However, our sensitivity tests indicate that the primary method and the SVD method give generally similar results. We choose the primary method for the main body of this paper because it ensures that we use exactly the same set of indices for each reanalysis. By contrast, the SVD method uses indices created from each reanalysis, which adds an additional layer of complexity to the intercomparison. Furthermore, we currently have no objective criteria for how many SVD cross-covariance components should

be considered (e.g. 10 versus 20) and, as discussed above, the SVD method carries an inherent risk of removing T2m variability that is associated with atmosphere-driven SST variability, and may therefore be part of the response to volcanic forcing.

**Appendix B: The warming over the NH continents in the first NH winter after the Pinatubo eruption**

The D91-JF92 anomalies associated with the Pinatubo eruption in Figure 3 show warming over Eurasia and cooling over much

of North America. As discussed in Section 4.1, the warming signals over Eurasia are in line with the results of previous studies, but the cooling signals over North America are not. We explore this discrepancy further in this section. Figure B1 shows the D91-JF92 anomalies obtained by using different sets of indices in the 1980–2010 MLR analysis based on JRA-55. Figure B1a is exactly the same as Figure 3a. Figures B1b–B1g results from "denial" studies where one or two tropical Pacific SST indices have been removed in the primary method. In Figure B1h, the simpler method employed by Kirchner et al. (1999, their Plate

1c) is applied; that is, we take anomalies that are "calculated with respect to a 15-year average over the Atmospheric Model Intercomparison Project (AMIP) time period of 1979–1993" for the same months of the year. Figure B1h agrees well with Plate 1c of Kirchner et al. (1999), including the strong warming signals over the southern and western parts of North America. We observe that the denial studies without Niño 3.4 (Fig. B1d), without Niño 3 (Fig. B1e), and without both Niño 3 and Niño 3.4 (Fig. B1g) show similar strong North American warming signals to Figure B1h and the previous studies. We have

conducted additional index-denial tests for other SST indices and found no essential differences in this region. The above results suggest that the warming signal over North America during this particular period as reported by previous studies is linked to combined SST variations in the tropical Pacific, and particularly the Niño 3 and Niño 3.4 regions, that are not fully considered in those analyses. The influence of Tropical Pacific SST variability onto North-American weather is achieved by the formation of a stationary Rossby wave forced by SST-modulated anomalies in tropical convective activity (Trenberth et

al., 1998). The propagation of this wave to the extratropics alters the atmospheric circulation in a way that resembles the Pacific/North American (PNA) teleconnection pattern (Barnston and Livezey, 1987; Straus and Shukla, 2002; Wallace and Gutzler, 1981) with a deep equivalent barotropic structure. The impacts of ENSO on the extratropical circulation and surface

temperatures over North America are known to differ between flavours of ENSO events (Garfinkel et al., 2013; Yu et al., 2012). Therefore, it is necessary to consider the whole spectrum of tropical Pacific variability to fully remove ENSO's impact on North American surface air temperature. Our results thus suggest that the D91-JF92 anomalies over most of North America following the Pinatubo eruption may actually have been stronger cooling in the northern and eastern parts of the continent and

weaker warming in the southern and western parts.

*Author contributions*

MF designed the study, made the MLR calculations, plotted the results, and drafted the original manuscript. PM proposed the use of EOF analysis and the SVD method, and made the EOF and SVD calculations. JSW provided information on the

reanalysis systems and on the tropical SST indices. All authors reviewed and edited the manuscript.

*Competing interests*

The authors declare that they have no conflict of interest.

*Special issue statement*

This article is part of the special issue "The SPARC Reanalysis Intercomparison Project (S-RIP) (ACP/ESSD inter-journal SI)". It is not associated with a conference.

*Acknowledgements*

We acknowledge the scientific guidance and sponsorship of the World Climate Research Programme (WCRP) coordinated in the framework of Stratosphere-troposphere Processes And their Role in Climate (SPARC) and SPARC Reanalysis Intercomparison Project (S-RIP). We thank the reanalysis centres for providing their support and data products. MF's contribution was financially supported in part by the Japan Society for the Promotion of Science (JSPS) through Grants-in-Aid for Scientific Research (JP26287117, JP16K05548, and JP18H01286). PM acknowledges support as an international

research fellow of the Japan Society for the Promotion of Science. The NXPACK library developed by Masato Shiotani was used for handling netCDF files by MF. The Linear Algebra PACKage (LAPACK) was used for the matrix operations. The

GFD-DENNOU library was used for producing Figures 1–9, A1–A3, and B1. We thank anonymous reviewers and Fanglin Yang for valuable comments and suggestions during the review process.

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

**Table 1. List of global atmospheric reanalysis data sets analysed in this study.**

| Name | Period covered | Grid spacing for forecast model | Reference |
|---|---|---|---|
| ERA-Interim | 1979–present | ~79 km | Dee et al. (2011) |
| ERA-40 | Sep. 1957–Aug. 2002 | ~125 km | Uppala et al. (2005) |
| ERA-20C | 1900–2010 | ~125 km | Poli et al. (2016) |
| CERA-20C | 1901–2010 | ~125 km | A website at the ECMWF [a] |
| JRA-55 | 1958–present | ~55 km | Kobayashi et al. (2015) |
| JRA-25 | Jan. 1979–Jan. 2014 | 1.125° | Onogi et al. (2007) |
| MERRA-2 | 1980–present | 0.5° latitude × 0.625° longitude | Gelaro et al. (2017) |
| MERRA | Jan. 1979–Feb. 2016 | 1/2° latitude × 2/3° longitude | Rienecker et al. (2011) |
| CFSR | Jan. 1979–Dec. 2010 | 0.3125° | Saha et al. (2010) |
| R-1 | 1948–present | 1.875° | Kalnay et al. (1996) |
| 20CRv2c | 1851–2014 | 1.875° | Compo et al. (2011) [b] |

[a] CERA-20C is a 10-member ensemble of coupled atmosphere-ocean-land-waves-sea ice reanalysis of the twentieth century, which assimilates only surface pressure and marine wind observations as well as ocean temperature and salinity profiles. More information can be found at https://www.ecmwf.int/en/forecasts/datasets/archive-datasets/reanalysis-datasets/cera-20c

5  (accessed 9 Nov. 2018). See also Laloyaux et al. (2016).

[b] Compo et al. (2011) and Fujiwara et al. (2017) described the 20CR version 2 (v2), which is a prior version of the 20CR version 2c (v2c) analysed in this study. The following website describes key updates and corrections for 20CRv2c relative to 20CRv2: https://www.esrl.noaa.gov/psd/data/gridded/data.20thC_ReanV2c.html (accessed 25 Feb. 2019).

**Table 2. Tropical and Arctic SST indices considered in this study. Indices are calculated as area-weighted averages of SST anomalies (SSTA) relative to the 1981–2010 base period in each specified region.**

|  | Region | Reference |
|---|---|---|
| Niño 1+2 | 90°W–80°W, Equator–10°S | Barnston et al. (1997) |
| Niño 3 | 150°W–90°W, 5°N–5°S | Barnston et al. (1997) |
| Niño 4 | 160°E–150°W, 5°N–5°S | Barnston et al. (1997) |
| Niño 3.4 | 170°W–120°W, 5°N–5°S | Barnston et al. (1997) |
| El Niño Modoki | SSTA (Region A) – 0.5 ×SSTA (Region B) – 0.5×SSTA (Region C) where<br>Region A: 165°E–140°W, 10°S–10°N<br>Region B: 110°W–70°W, 15°S–5°N<br>Region C: 125°E–145°E, 10°S–20°N | Ashok et al. (2007) |
| Indian Ocean basin mode | 40°E–100°E, 20°S–20°N | Zheng et al. (2011) |
| Indian Ocean dipole mode | SSTA (Region W) – SSTA (Region E) where<br>Region W: 50°E–70°E, 10°S–10°N<br>Region E: 90°E–110°E, 10°S–Equator | Saji et al. (1999) |
| Atlantic cold tongue (or Atlantic Niño) | 15°W–5°W, 3°S–3°N | Richter et al. (2013) |
| northern tropical Atlantic | 40°W–10°W, 10°N–20°N | Richter et al. (2013) |
| Barents-Kara Sea region | 30°E–70°E, 70°N–80°N [a] | Kug et al. (2015) |
| Chukchi Sea region | 160°E–160°W, 65°N–80°N [a] | Kug et al. (2015) |

[a] Land regions are masked when calculating regional means.

**Table 3. Monthly mean 2-metre temperature data from global atmospheric reanalysis data sets.**

| Name | Uniform resource locator (URL) or digital object identifier (DOI) | Date accessed |
|---|---|---|
| ERA-Interim | http://apps.ecmwf.int/datasets/data/interim-full-moda/levtype=sfc/ (0.75° × 0.75° grid) | 19 Apr. 2017 |
| ERA-40 | http://apps.ecmwf.int/datasets/data/era40-moda/levtype=sfc/ (1° × 1° grid) | 19 Apr. 2017 |
| ERA-20C | http://apps.ecmwf.int/datasets/data/era20c-moda/levtype=sfc/type=an/ (1° × 1° grid) | 19 Apr. 2017 |
| CERA-20C | http://apps.ecmwf.int/datasets/data/cera20c-edmo/levtype=sfc/type=an/ (1° × 1° grid) | 19 Apr. 2017 |
| JRA-55 | ftp://ds.data.jma.go.jp | 13–14 May 2015 |
| JRA-25 | http://jra.kishou.go.jp/ (Not available now; access https://rda.ucar.edu/datasets/ds625.1/) | 23–24 Mar. 2012 |
| MERRA-2 | https://doi.org/10.5067/AP1B0BA5PD2K ("tavgM_2d_slv_Nx" files; GMAO, 2015) | 6 Apr. 2017 |
| MERRA | http://disc.sci.gsfc.nasa.gov/daac-bin/FTPSubset.pl?LOOKUPID_List=MATMNXSLV ("tavgM_2d_slv_Nx" files) | 6 Apr. 2017 |
| CFSR | https://nomads.ncdc.noaa.gov/data/cfsrmon/ ("flxf06" files) | 10 Jul. 2018 |
| R-1 | https://www.esrl.noaa.gov/psd/data/gridded/data.ncep.reanalysis.derived.html (Surface Fluxes) | 6 Dec. 2017 |
| 20CRv2c | https://www.esrl.noaa.gov/psd/data/gridded/data.20thC_ReanV2c.html (Monthly, Single level) | 3 Apr. 2017 |

**Table 4. Monthly mean data sets for climatic indices.**

| Name | URL | Date accessed |
|---|---|---|
| ERSSTv5 | https://www.esrl.noaa.gov/psd/data/gridded/data.noaa.ersst.v5.html | 13–14 Dec. 2017 |
| Monthly Niño indices | http://www.cpc.ncep.noaa.gov/data/indices/ (Monthly ERSSTv5 (1981-2010 base period)) | 6 Jun. 2018 |
| QBO indices | http://www.geo.fu-berlin.de/en/met/ag/strat/produkte/qbo/ | 6 Jun. 2018 |
| Solar flux | ftp://ftp.geolab.nrcan.gc.ca/data/solar_flux/ (through https://www.ngdc.noaa.gov/stp/solar/flux.html) | 6 Jun. 2018 |

Figure captions:

**Figure 1. (a) Time series of the temperature residual $R(t)$ (including volcanic signals and random variations) averaged for (a) 60°N–60°S and (b) 60°N–Equator as obtained from the 1980–2010 MLR analysis for 10 reanalysis data sets as well as reanalysis ensemble mean (REM) (see legend at top). Three-month running means have been applied to each time series. (c) Time series of aerosol optical depth at 550 nm in the stratosphere (Sato et al., 1993; obtained from https://data.giss.nasa.gov/modelforce/strataer/ (25 July 2018); black solid for the global mean, dashed darker grey for the NH mean, and dashed lighter grey for the SH mean). For all panels, vertical dashed lines indicate the start dates of the two volcanic eruptions.**

**Figure 2. Geographical and zonal-mean distributions of the 2-metre temperature anomalies averaged for September to November 1992 following the Mount Pinatubo eruption in June 1991. Results are based on the 1980–2010 MLR analyses for each of 10 reanalysis data sets (see the legend at the top of each panel). Solid and dashed contours denote positive and negative anomalies, respectively. The contour interval is 0.5 K, without 0.0 K lines. Coloured shading denotes anomalies that are positive (orange) or negative (blue) with magnitudes exceeding one standard deviation (SD) of the 3-month mean $R(t)$ at that location.**

**Figure 3. As for Figure 2, but for the anomalies averaged from December 1991 to February 1992 (top) and from June to August 1992 (bottom) following the Mount Pinatubo eruption as calculated using JRA-55 (left) and R-1 (right). The contour interval is 0.5 K, without 0.0 K lines. Coloured shading denotes anomalies that are positive (orange) or negative (blue) with magnitudes exceeding one SD of the 3-month mean $R(t)$ at that location.**

**Figure 4. As for Figure 2, but for the anomalies averaged over June to August 1983 following the El Chichón eruption in April 1982. The contour interval is 0.5 K, without 0.0 K lines. Coloured shading denotes anomalies that are positive (orange) or negative (blue) with magnitudes exceeding one SD of the 3-month mean $R(t)$ at that location.**

**Figure 5. As for Figure 1, but based on the 1958–2001 MLR analyses for six reanalysis data sets; averages for Equator–60°S are shown in panel (b). For all panels, vertical dotted lines indicate the starting date of the three volcanic eruptions.**

**Figure 6. As for Figure 2, but based on the 1958–2001 MLR analysis for each of six reanalysis data sets. The contour interval is 0.5 K, without 0.0 K lines. Coloured shading denotes anomalies that are positive (orange) or negative (blue) with magnitudes exceeding one SD of the 3-month mean $R(t)$ at that location.**

**Figure 7. As for Figure 4, but based on the 1958–2001 MLR analysis for each of six reanalysis data sets. The contour interval is 0.5 K, without 0.0 K lines. Coloured shading denotes anomalies that are positive (orange) or negative (blue) with magnitudes exceeding one SD of the 3-month mean $R(t)$ at that location.**

**Figure 8.** As for Figure 7, but for the anomalies averaged over June to August 1964 following the Mount Agung eruption in March 1963. The contour interval is 0.5 K, without 0.0 K lines. Coloured shading denotes anomalies that are positive (orange) or negative (blue) with magnitudes exceeding one SD of the 3-month mean $R(t)$ at that location.

**Figure 9.** Inter-reanalysis differences presented in standard deviation (SD) for (a) September to November 1992 and (b) December 1991 to February 1992 (both following the Mount Pinatubo eruption) based on the 1980–2010 MLR analysis with 10 reanalysis data sets, and for (c) September to November 1992 (following the Mount Pinatubo eruption) and (d) June to August 1964 (following the Mount Agung eruption) based on the 1958–2001 MLR analysis with six reanalysis data sets. The 3-month average for each reanalysis data set has been re-gridded to a 2.5°×2.5° grid (if necessary) before calculating the SD. The contour interval is 0.2 K. Regions with SD values exceeding 0.2 K are coloured green, with a light shade for 0.2–0.4 K, a darker shade for 0.4–0.6 K, and the darkest shade for 0.6 K and greater. Zonal means for each of the four cases are also shown.

**Figure A1.** As for Figure 2, but comparing different methods to obtain the 2-metre temperature anomalies in SON 1992 following the Mount Pinatubo eruption. Only the JRA-55 reanalysis data set is used, with (a) the primary method, as in Figure 2a, (b) the SVD method with the first 10 cross-covariance components, (c) the single-Niño method with the Niño 1+2 index, (d) the single-Niño method with the Niño 4 index, and (e) the single-Niño method with the Niño 3.4 index. See text for the details of each method. The contour interval is 0.5 K, without 0.0 K lines. Coloured shading denotes that anomalies are positive (orange) or negative (blue) with absolute magnitudes larger than one SD of the 3-month mean $R(t)$ at that location.

**Figure A2.** As for Figure A1, but for the JJA 1983 anomalies following the El Chichón eruption in April 1982. The contour interval is 0.5 K, without 0.0 K lines. Coloured shading denotes anomalies that are positive (orange) or negative (blue) with magnitudes exceeding one SD of the 3-month mean $R(t)$ at that location.

**Figure A3.** As for Figure 1, but showing results for the primary method (i.e. as shown in Figure 1; gray) versus results for the SVD method with the first 10 cross-covariance components (black) using JRA-55 data.

**Figure B1.** Similar to Figure A1, but for the anomalies averaged from December 1991 to February 1992 following the Mount Pinatubo eruption using JRA-55, and with changes to panels (b)–(e) and with additions of panels (f)–(h). (a) The primary result as in Figure 3a, (b) the result using the primary method but without the El Niño Modoki index, (c) the result using the primary method but without the Niño 1+2 index, (d) the result using the primary method but without the Niño 3.4 index, (e) the result using the primary method but without the Niño 3 index, (f) the result using the primary method but without the Niño 4 index, (g) the result using the primary method but without the Niño 3 and Niño 3.4 indices, and (h) the anomalies with respect to the 1979–1993 means (for the same months of year) following Plate 1c of Kirchner et al. (1999). See text for the details of each method. The contour interval is 0.5 K, without 0.0 K lines. Coloured shading in (a)–(g) denotes anomalies that are positive (orange) or negative (blue) with absolute magnitudes larger than one SD of the 3-month mean $R(t)$. Coloured shading in (h) has a similar meaning but with anomalies evaluated against the SD of DJF-mean data during 1979–1993.

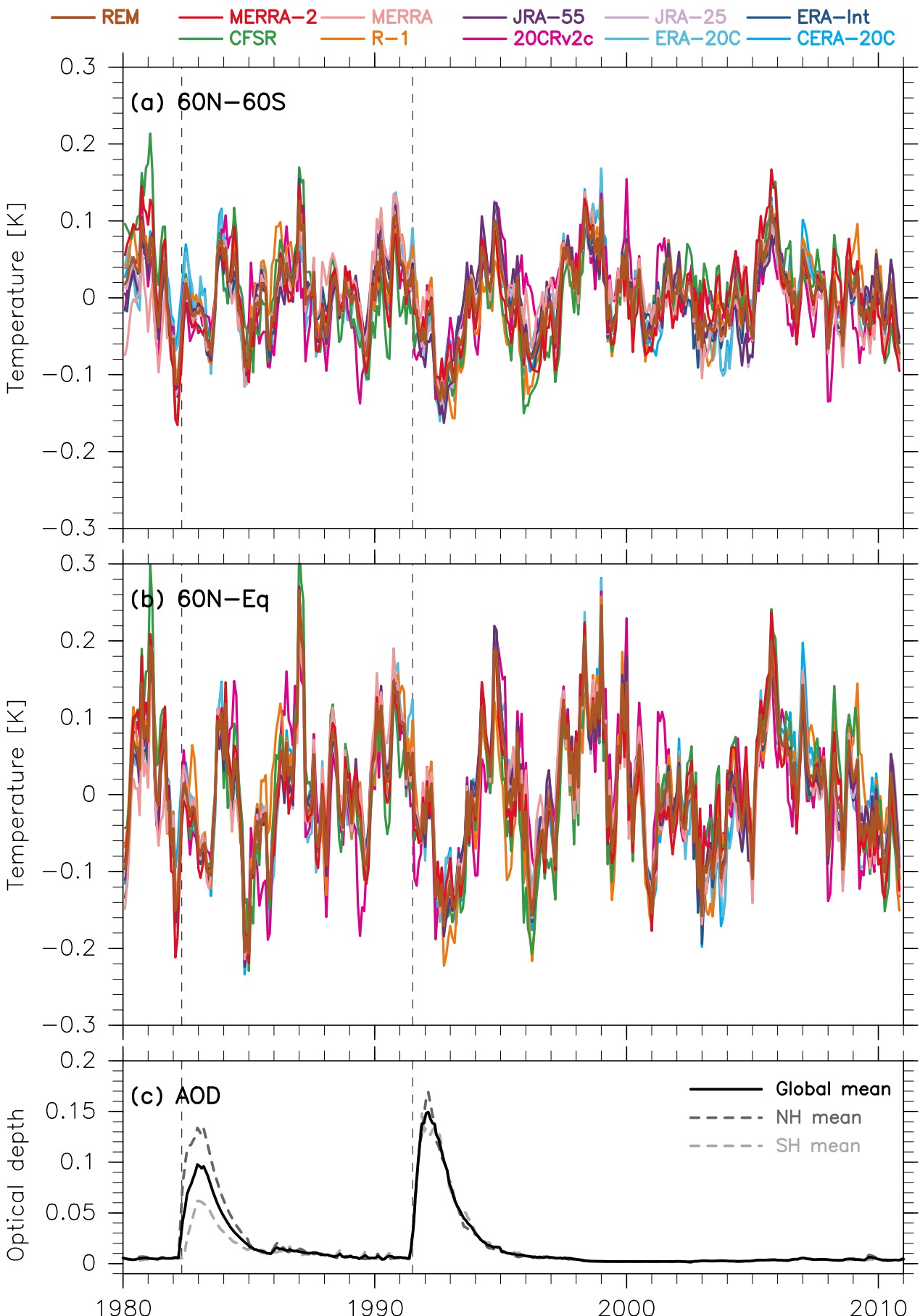

Figure 1. (a) Time series of the temperature residual R(t) (including volcanic signals and random variations) averaged for (a) 60°N–60°S and (b) 60°N–Equator as obtained from the 1980–2010 MLR analysis for 10 reanalysis data sets as well as reanalysis ensemble mean (REM) (see legend at top). Three-month running means have been applied to each time series. (c) Time series of aerosol optical depth at 550 nm in the stratosphere (Sato et al., 1993; obtained from https://data.giss.nasa.gov/modelforce/strataer/ (25 July 2018); black solid for the global mean, dashed darker grey for the NH mean, and dashed lighter grey for the SH mean). For all panels, vertical dashed lines indicate the start dates of the two volcanic eruptions.

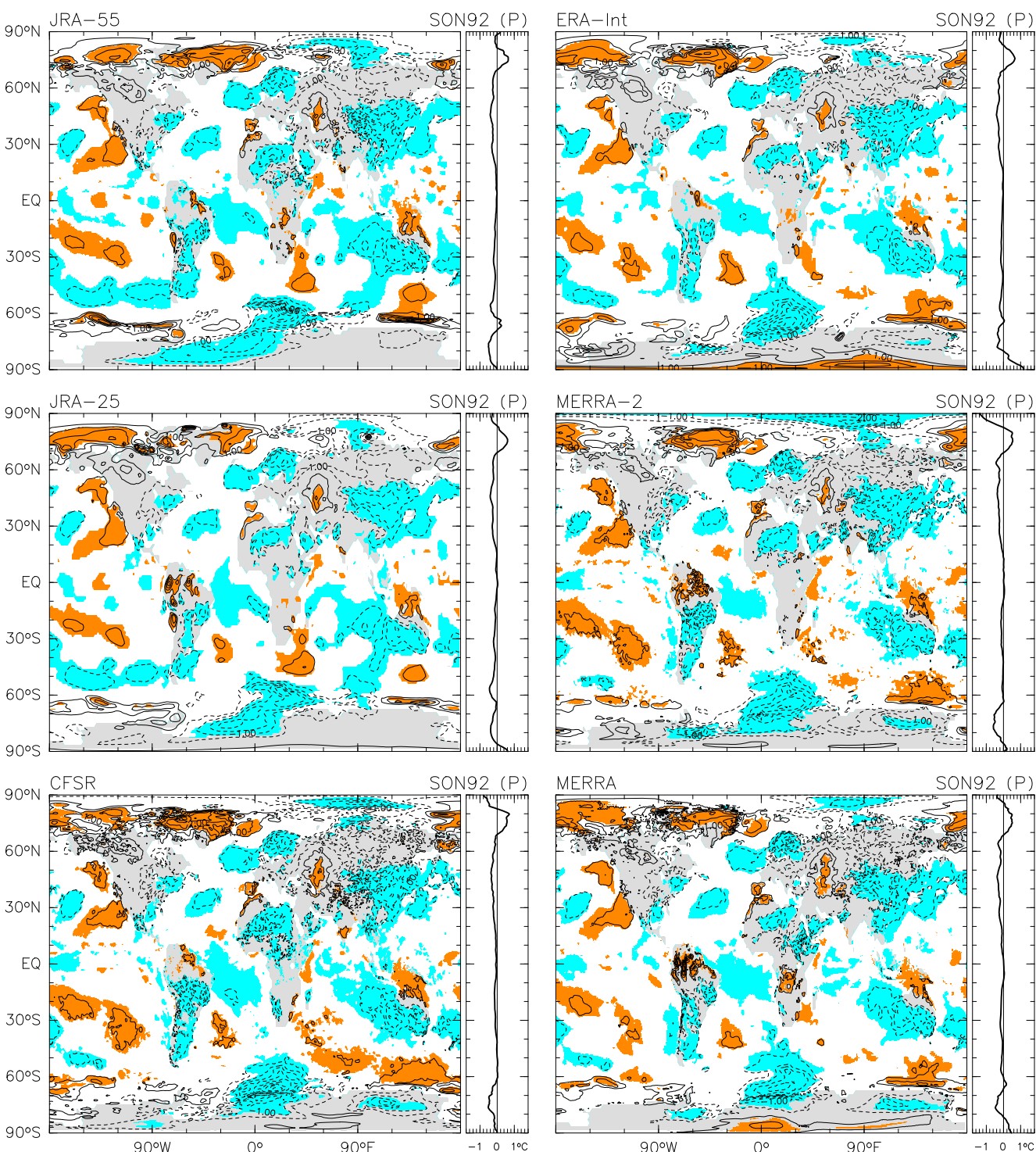

Figure 2. Geographical and zonal-mean distributions of the 2-metre temperature anomalies averaged for September to November 1992 following the Mount Pinatubo eruption in June 1991. Results are based on the 1980–2010 MLR analyses for each of 10 reanalysis data sets (see the legend at the top of each panel). Solid and dashed contours denote positive and negative anomalies, respectively. The contour interval is 0.5 K, without 0.0 K lines. Coloured shading denotes anomalies that are positive (orange) or negative (blue) with magnitudes exceeding one standard deviation (SD) of the 3-month mean R(t) at that location.

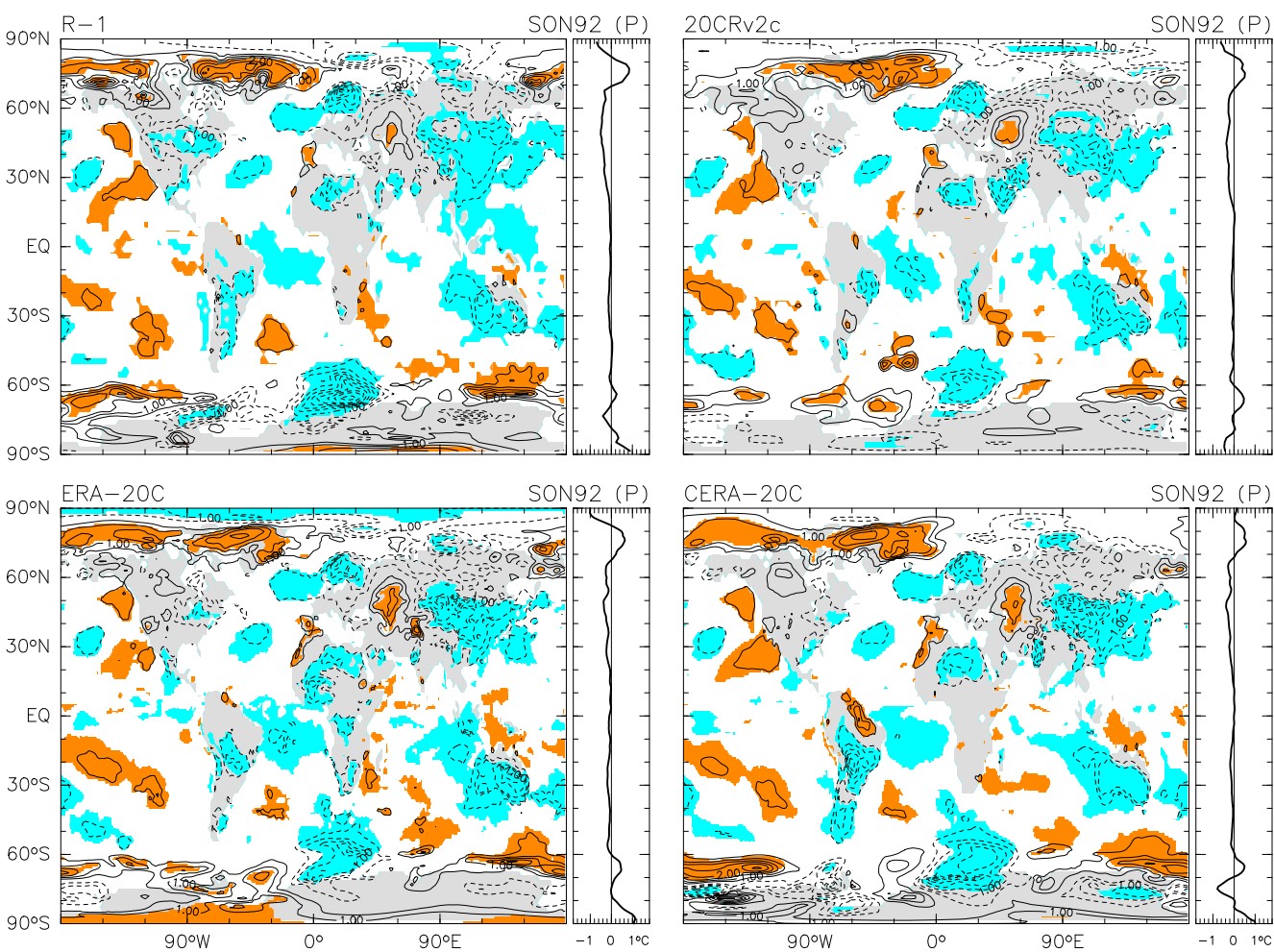

Figure 2. (continued)

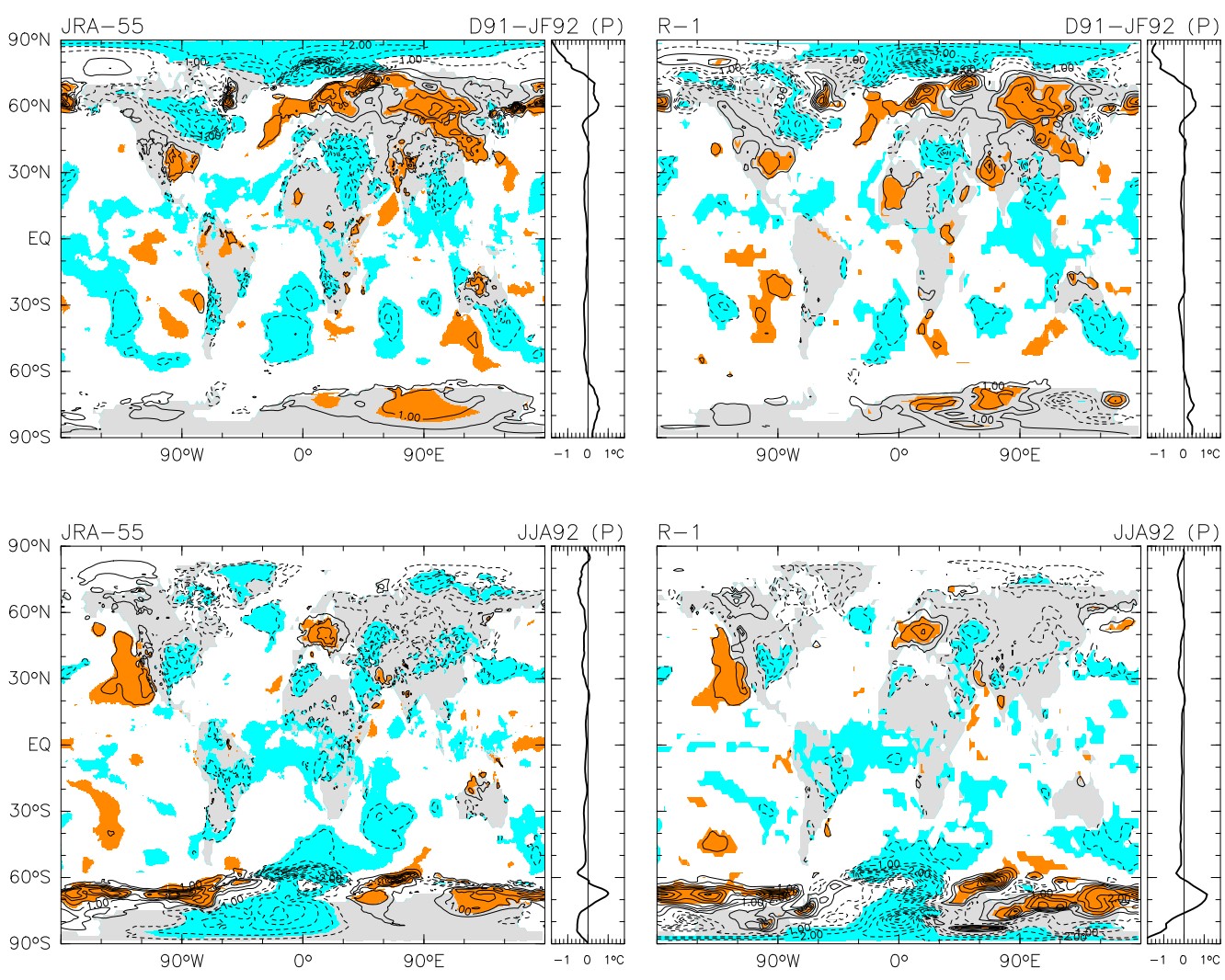

Figure 3. As for Figure 2, but for the anomalies averaged from December 1991 to February 1992 (top) and from June to August 1992 (bottom) following the Mount Pinatubo eruption as calculated using JRA-55 (left) and R-1 (right). The contour interval is 0.5 K, without 0.0 K lines. Coloured shading denotes anomalies that are positive (orange) or negative (blue) with magnitudes exceeding one SD of the 3-month mean R(t) at that location.

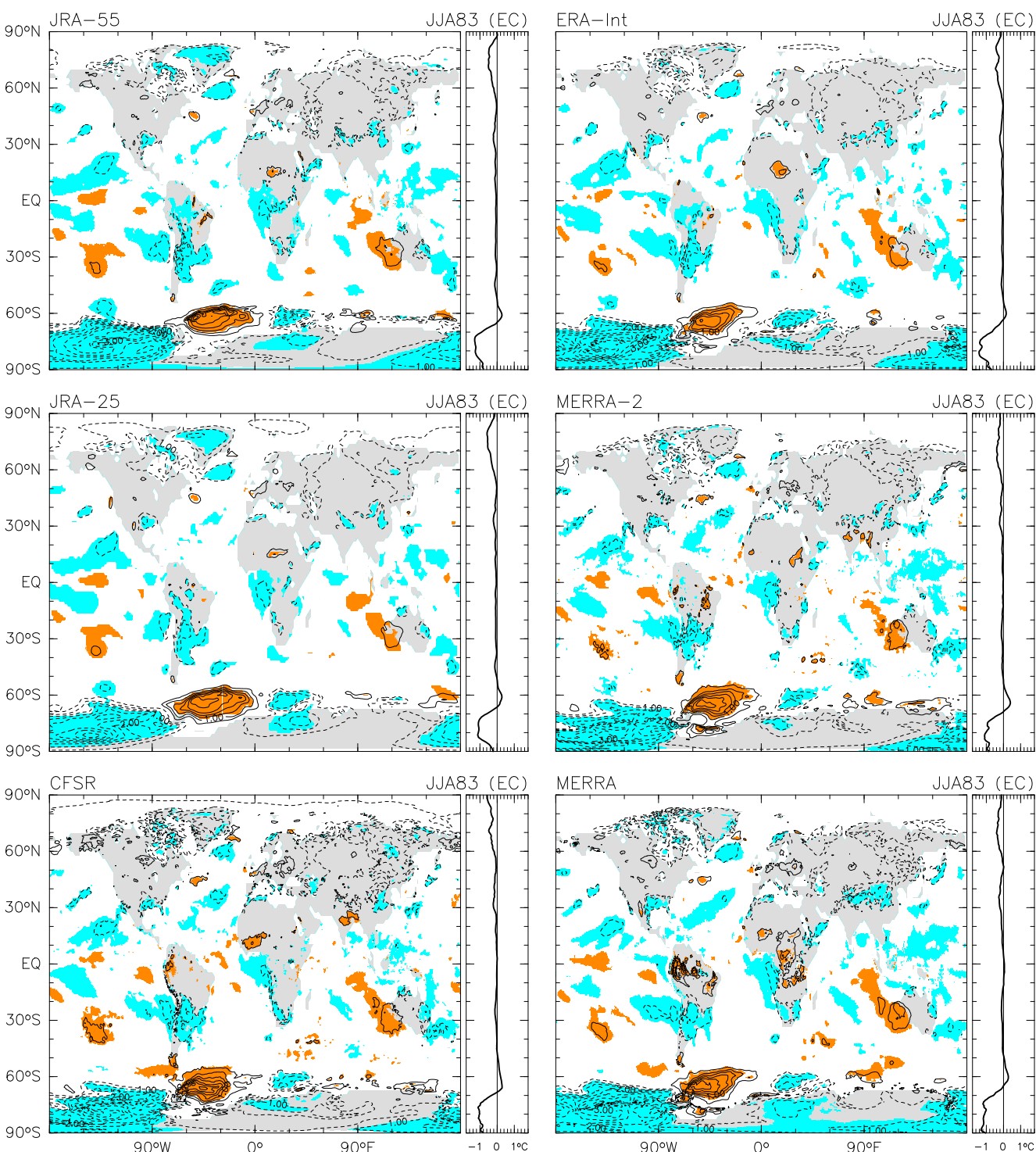

Figure 4. As for Figure 2, but for the anomalies averaged over June to August 1983 following the El Chichón eruption in April 1982. The contour interval is 0.5 K, without 0.0 K lines. Coloured shading denotes anomalies that are positive (orange) or negative (blue) with magnitudes exceeding one SD of the 3-month mean R(t) at that location.

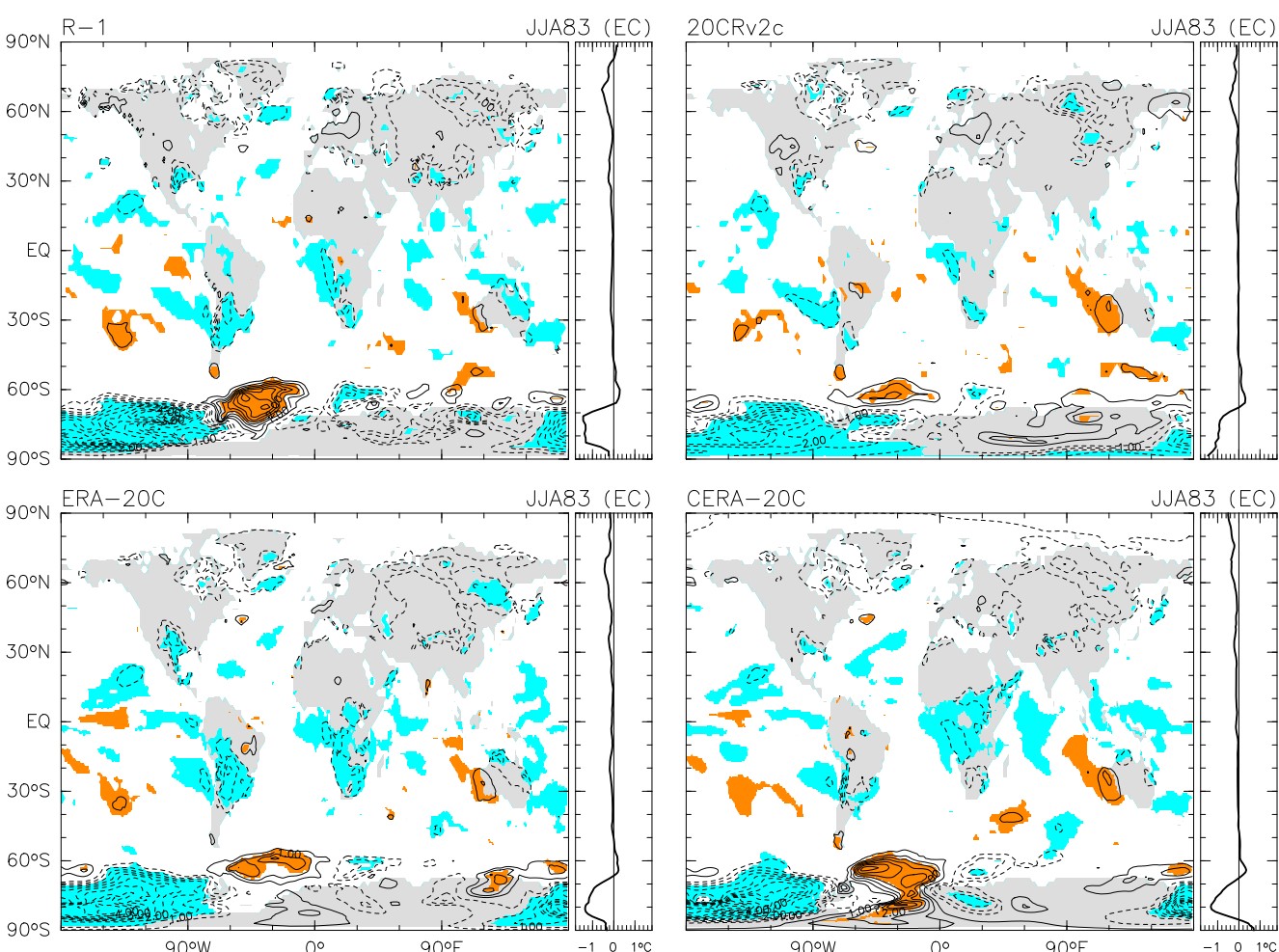

Figure 4. (continued)

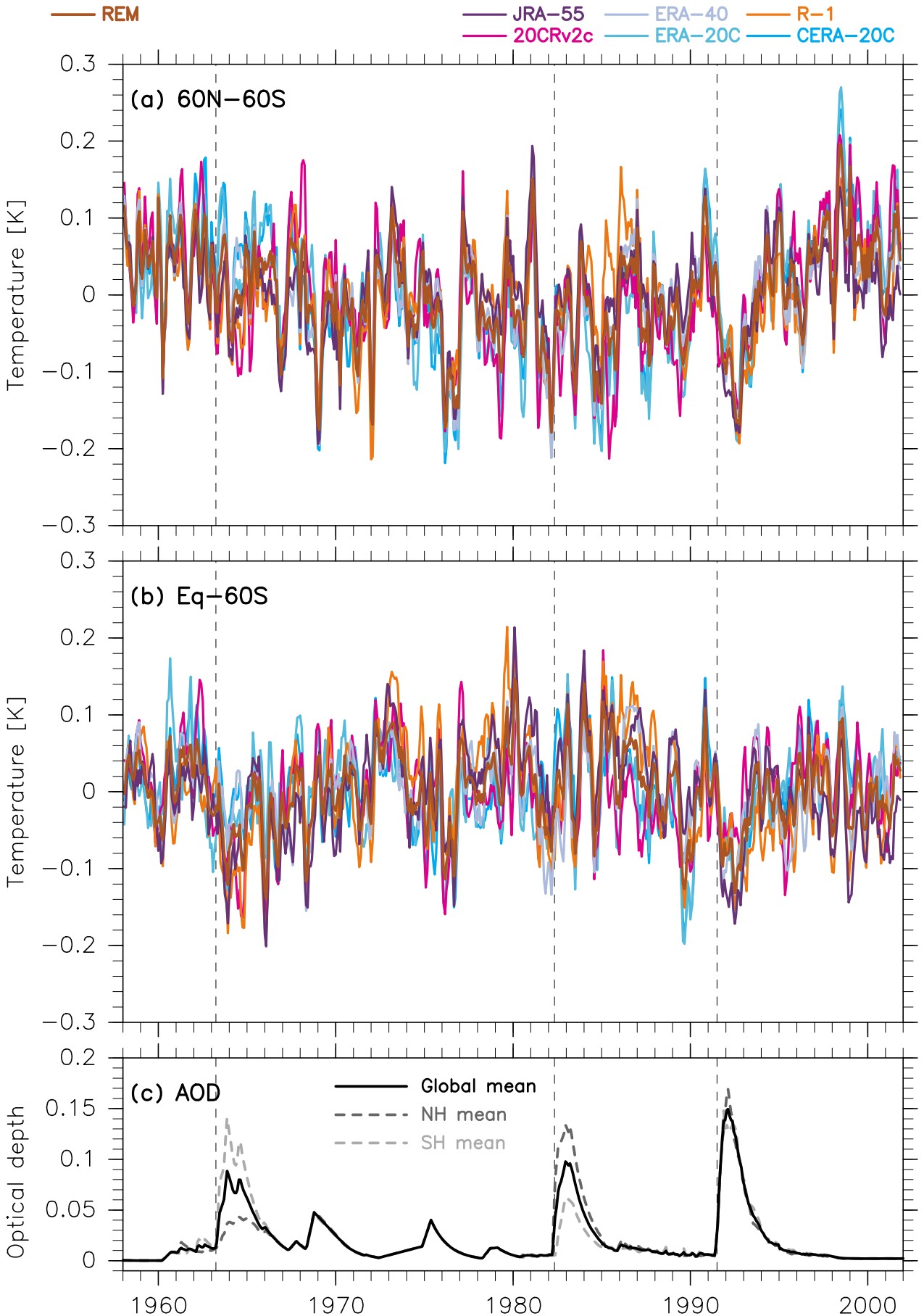

Figure 5. As for Figure 1, but based on the 1958–2001 MLR analyses for six reanalysis data sets; averages for Equator–60°S are shown in panel (b). For all panels, vertical dotted lines indicate the starting date of the three volcanic eruptions.

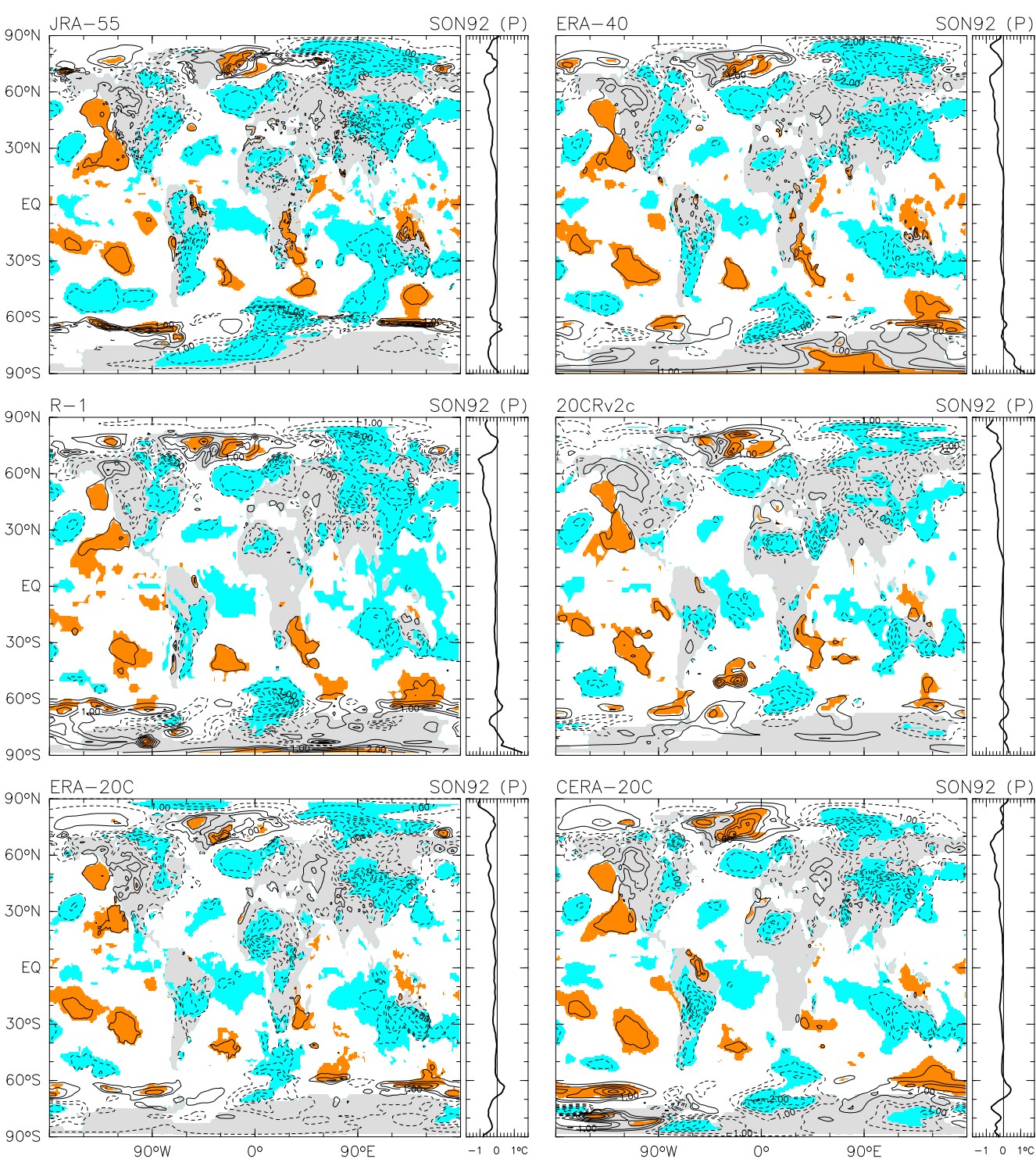

Figure 6. As for Figure 2, but based on the 1958–2001 MLR analysis for each of six reanalysis data sets. The contour interval is 0.5 K, without 0.0 K lines. Coloured shading denotes anomalies that are positive (orange) or negative (blue) with magnitudes exceeding one SD of the 3-month mean R(t) at that location.

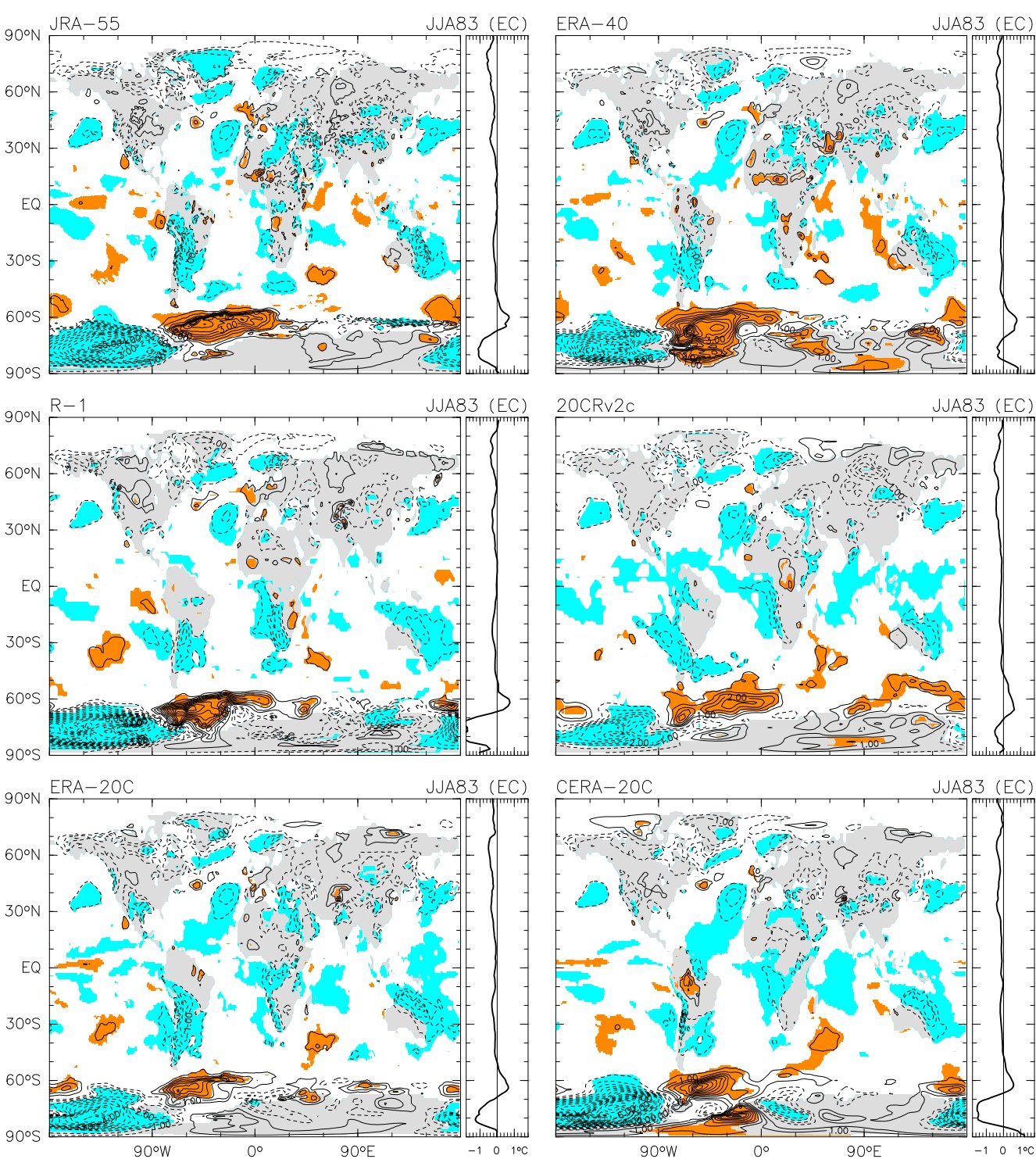

Figure 7. As for Figure 4, but based on the 1958–2001 MLR analysis for each of six reanalysis data sets. The contour interval is 0.5 K, without 0.0 K lines. Coloured shading denotes anomalies that are positive (orange) or negative (blue) with magnitudes exceeding one SD of the 3-month mean R(t) at that location.

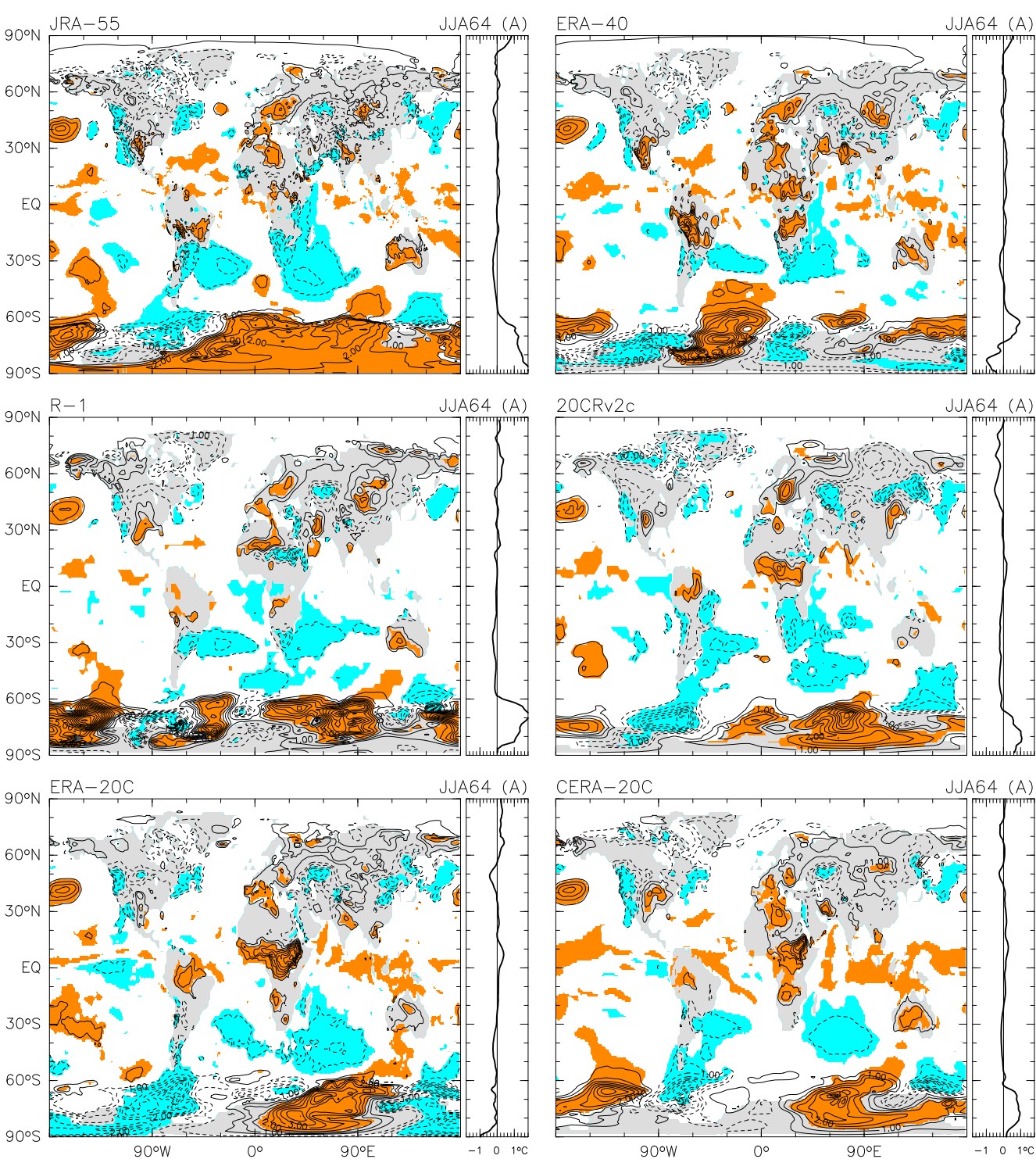

Figure 8. As for Figure 7, but for the anomalies averaged over June to August 1964 following the Mount Agung eruption in March 1963. The contour interval is 0.5 K, without 0.0 K lines. Coloured shading denotes anomalies that are positive (orange) or negative (blue) with magnitudes exceeding one SD of the 3-month mean R(t) at that location.

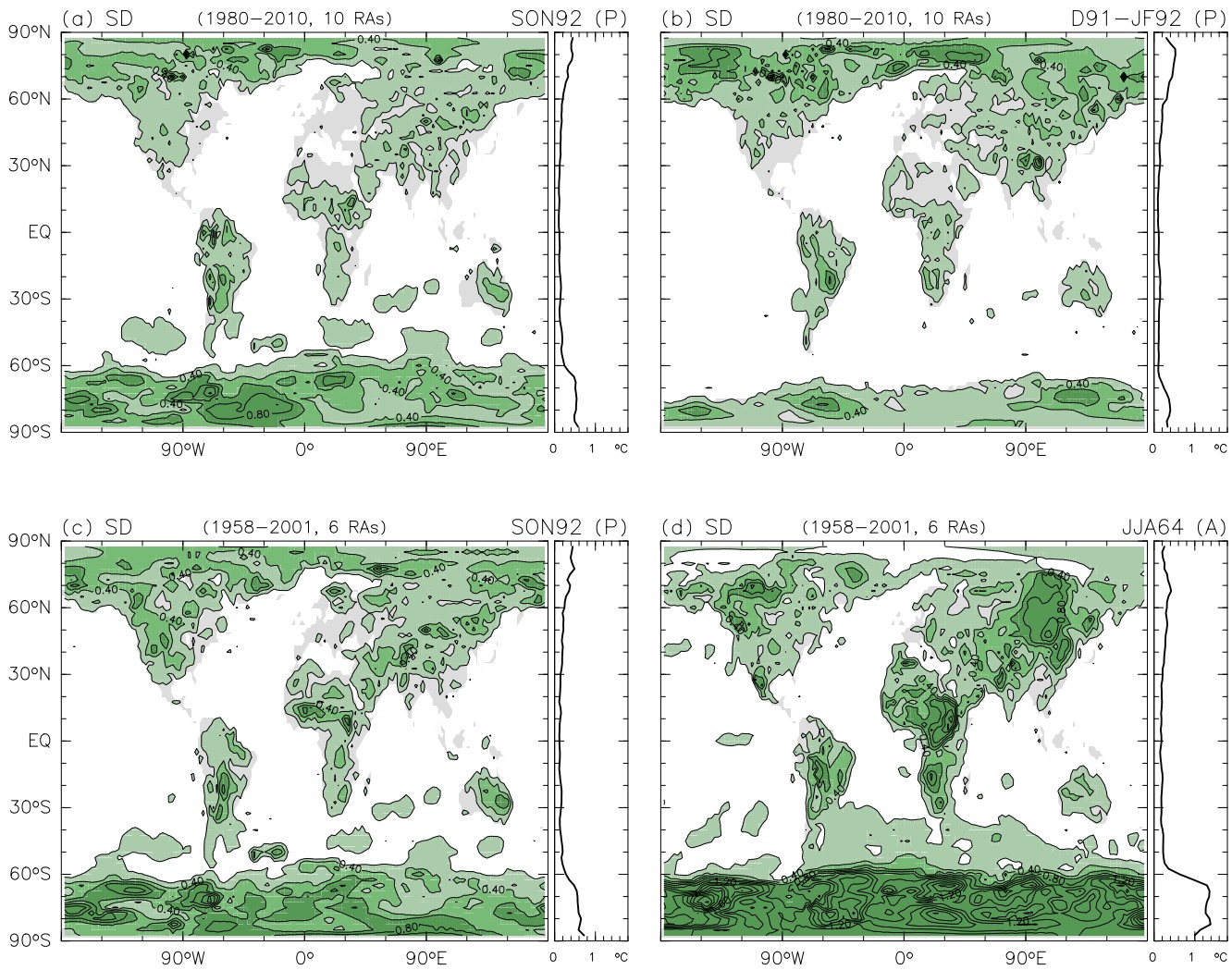

Figure 9. Inter-reanalysis differences presented in standard deviation (SD) for (a) September to November 1992 and (b) December 1991 to February 1992 (both following the Mount Pinatubo eruption) based on the 1980–2010 MLR analysis with 10 reanalysis data sets, and for (c) September to November 1992 (following the Mount Pinatubo eruption) and (d) June to August 1964 (following the Mount Agung eruption) based on the 1958–2001 MLR analysis with six reanalysis data sets. The 3-month average for each reanalysis data set has been re-gridded to a 2.5°×2.5° grid (if necessary) before calculating the SD. The contour interval is 0.2 K. Regions with SD values exceeding 0.2 K are coloured green, with a light shade for 0.2–0.4 K, a darker shade for 0.4–0.6 K, and the darkest shade for 0.6 K and greater. Zonal means for each of the four cases are also shown.

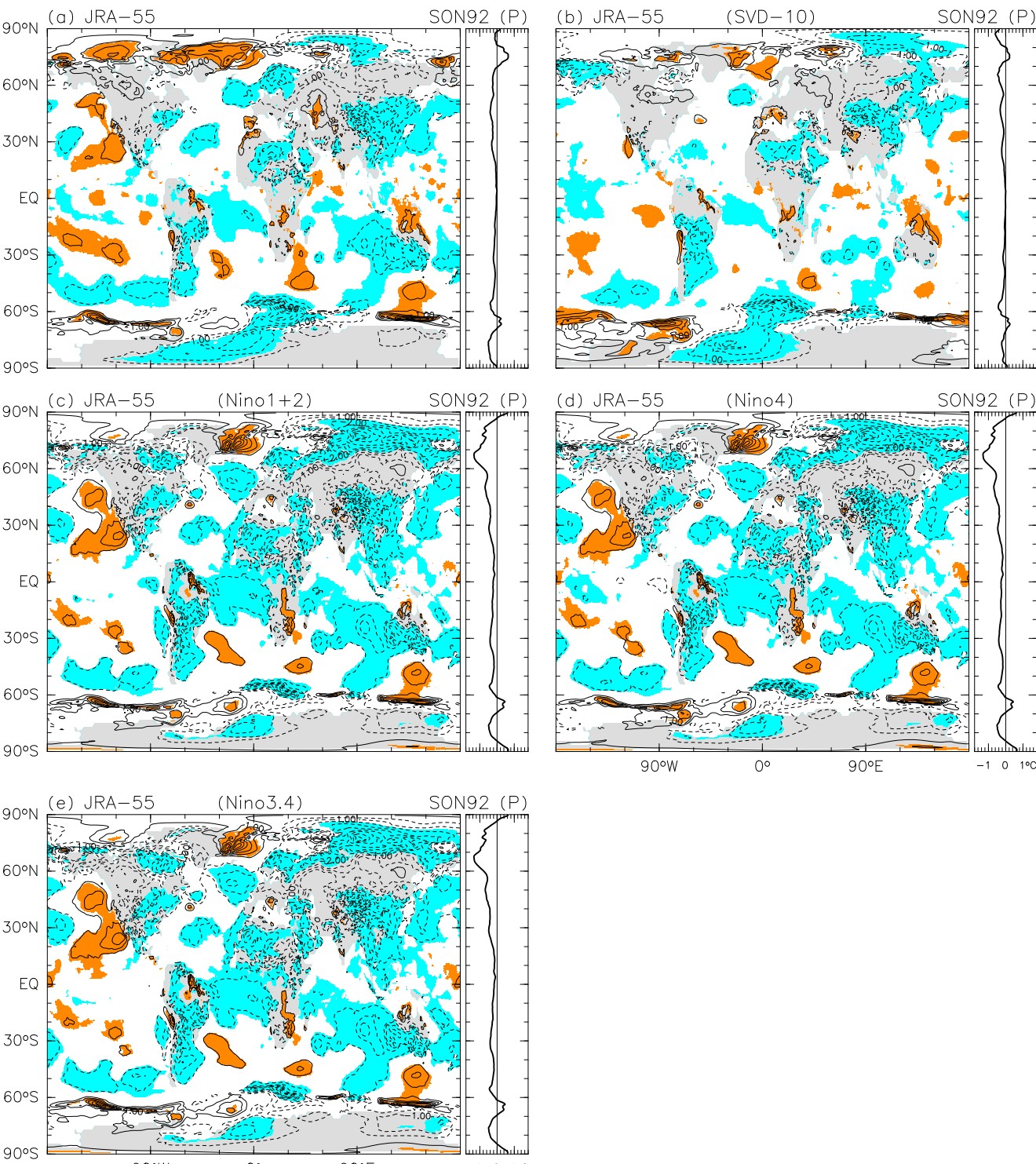

Figure A1. As for Figure 2, but comparing different methods to obtain the 2-metre temperature anomalies in SON 1992 following the Mount Pinatubo eruption. Only the JRA-55 reanalysis data set is used, with (a) the primary method, as in Figure 2a, (b) the SVD method with the first 10 cross-covariance components, (c) the single-Niño method with the Niño 1+2 index, (d) the single-Niño method with the Niño 4 index, and (e) the single-Niño method with the Niño 3.4 index. See text for the details of each method. The contour interval is 0.5 K, without 0.0 K lines. Coloured shading denotes that anomalies are positive (orange) or negative (blue) with absolute magnitudes larger than one SD of the 3-month mean R(t) at that location.

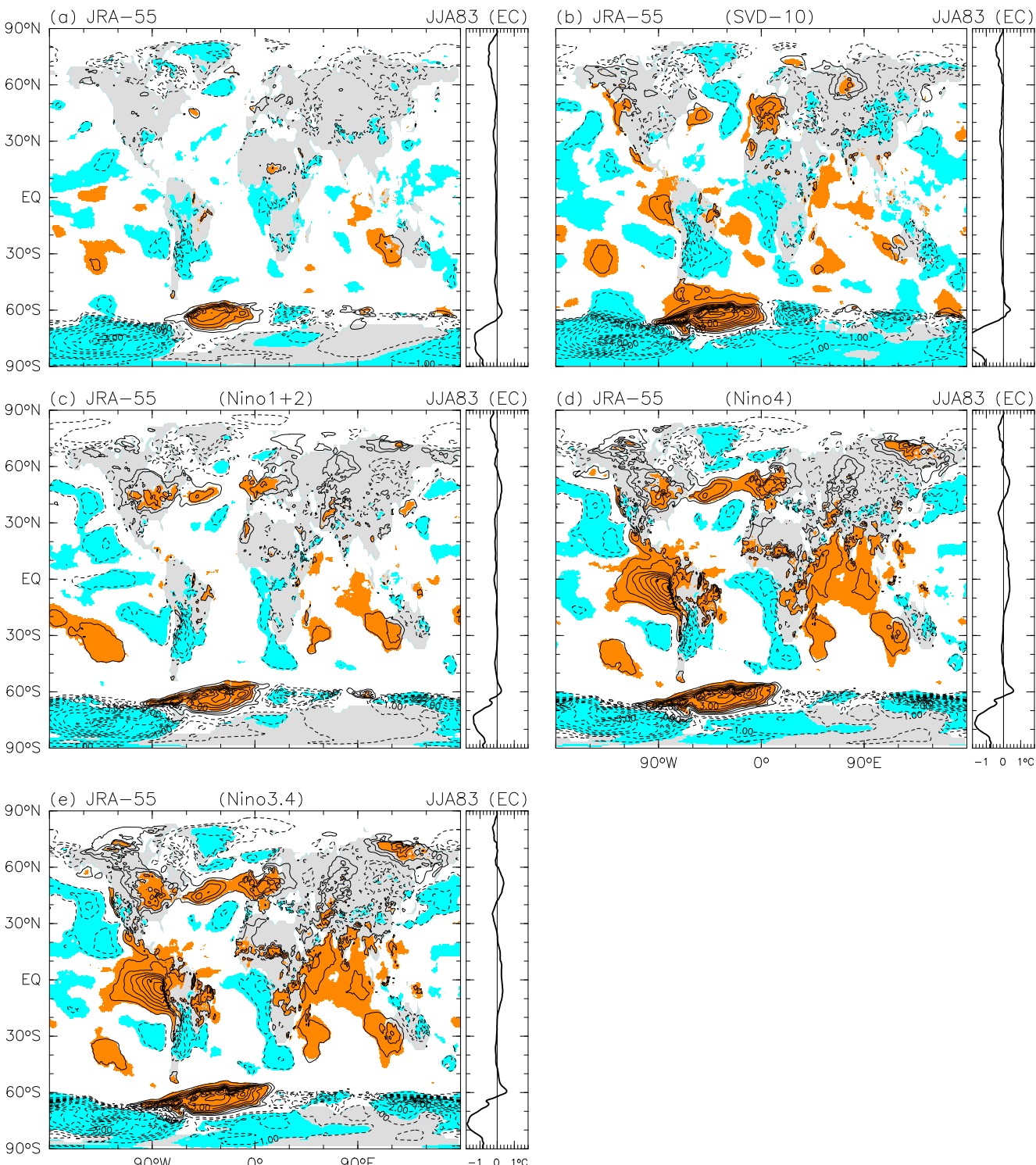

Figure A2. As for Figure A1, but for the JJA 1983 anomalies following the El Chichón eruption in April 1982. The contour interval is 0.5 K, without 0.0 K lines. Coloured shading denotes anomalies that are positive (orange) or negative (blue) with magnitudes exceeding one SD of the 3-month mean R(t) at that location.

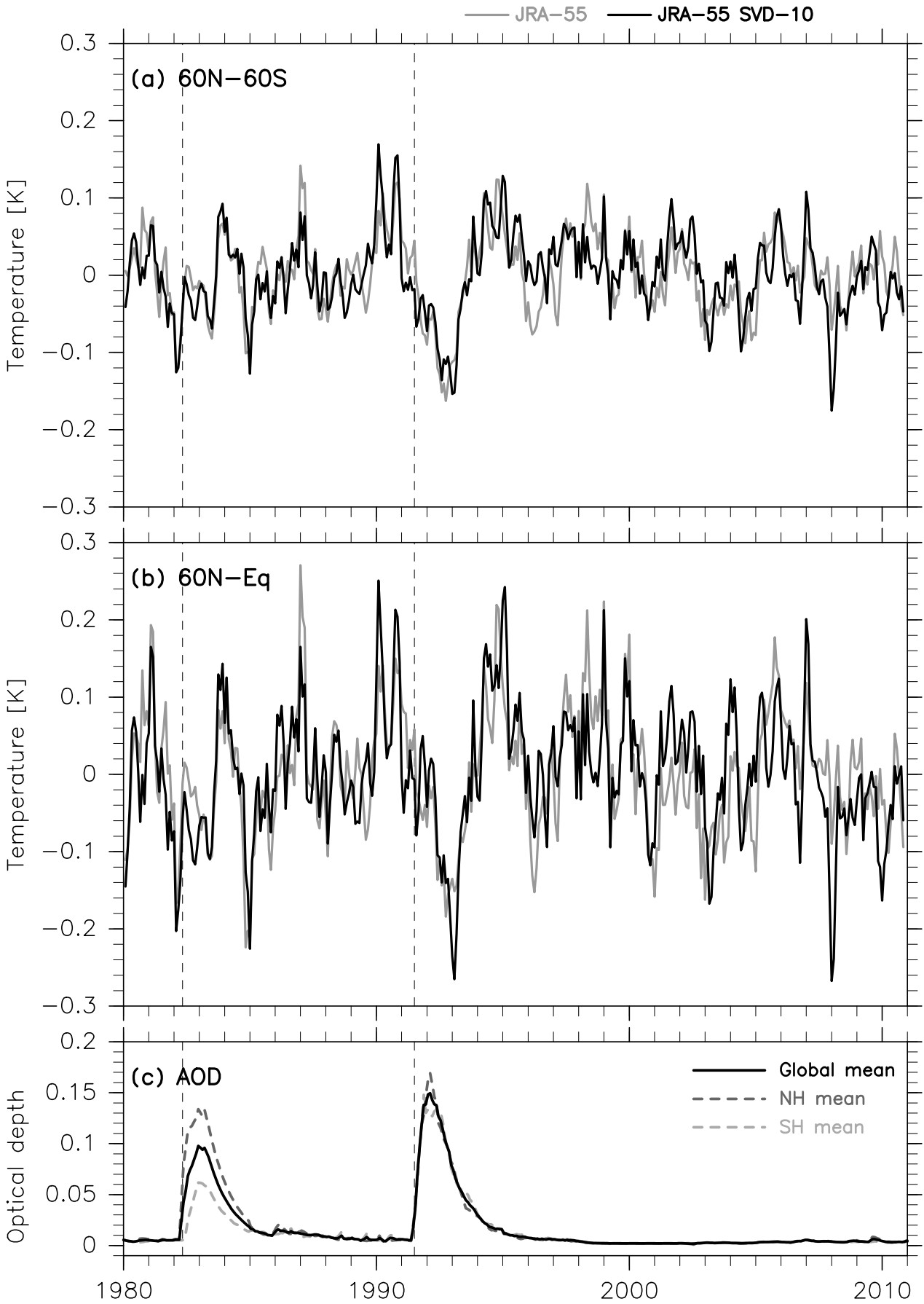

Figure A3. As for Figure 1, but showing results for the primary method (i.e. as shown in Figure 1; gray) versus results for the SVD method with the first 10 cross-covariance components (black) using JRA-55 data.

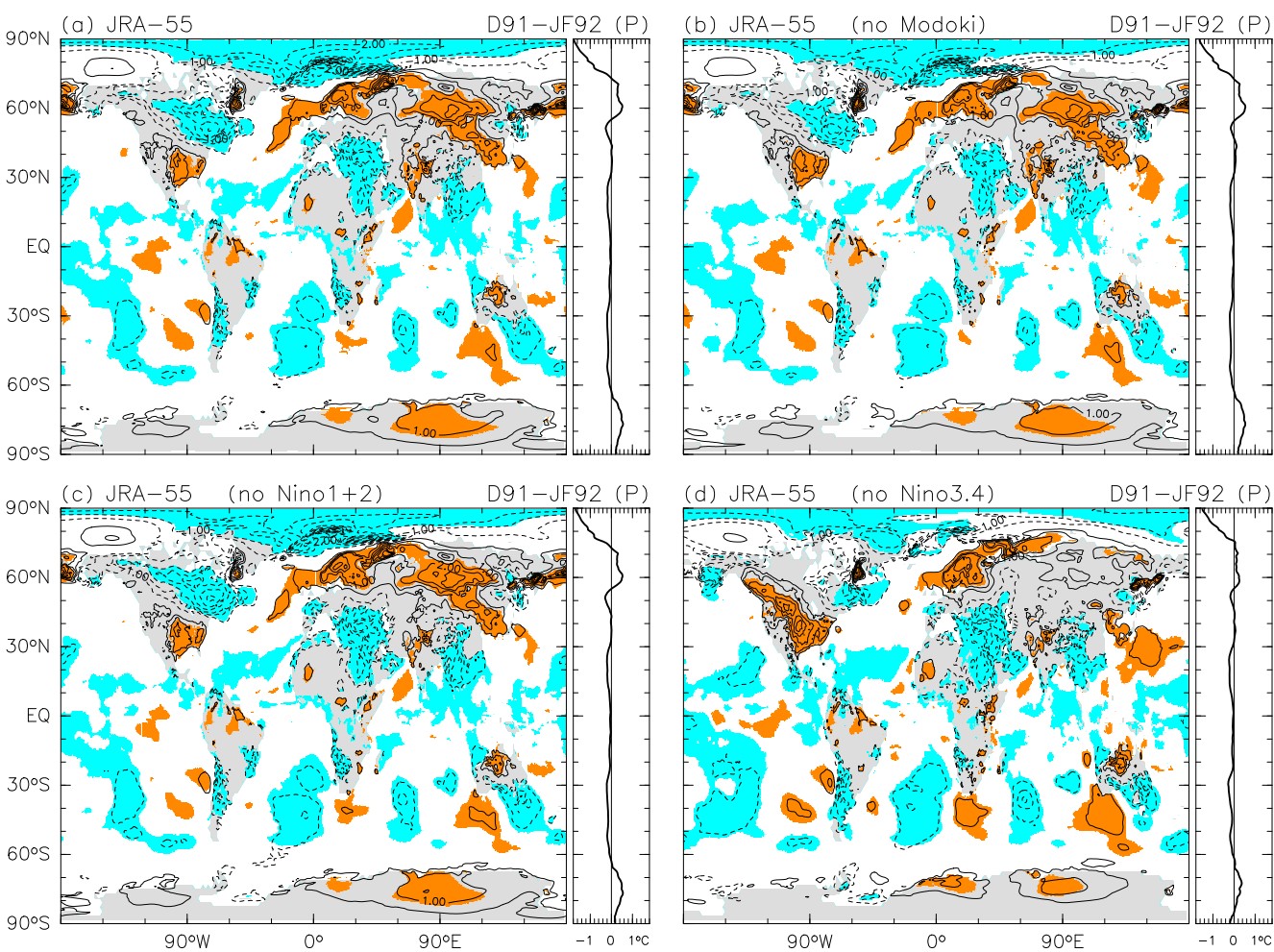

Figure B1. Similar to Figure A1, but for the anomalies averaged from December 1991 to February 1992 following the Mount Pinatubo eruption using JRA-55, and with changes to panels (b)–(e) and with additions of panels (f)–(h). (a) The primary result as in Figure 3a, (b) the result using the primary method but without the El Niño Modoki index, (c) the result using the primary method but without the Niño 1+2 index, (d) the result using the primary method but without the Niño 3.4 index, (e) the result using the primary method but without the Niño 3 index, (f) the result using the primary method but without the Niño 4 index, (g) the result using the primary method but without the Niño 3 and Niño 3.4 indices, and (h) the anomalies with respect to the 1979–1993 means (for the same months of year) following Plate 1c of Kirchner et al. (1999). See text for the details of each method. The contour interval is 0.5 K, without 0.0 K lines. Coloured shading in (a)–(g) denotes anomalies that are positive (orange) or negative (blue) with absolute magnitudes larger than one SD of the 3-month mean R(t). Coloured shading in (h) has a similar meaning but with anomalies evaluated against the SD of DJF-mean data during 1979–1993.

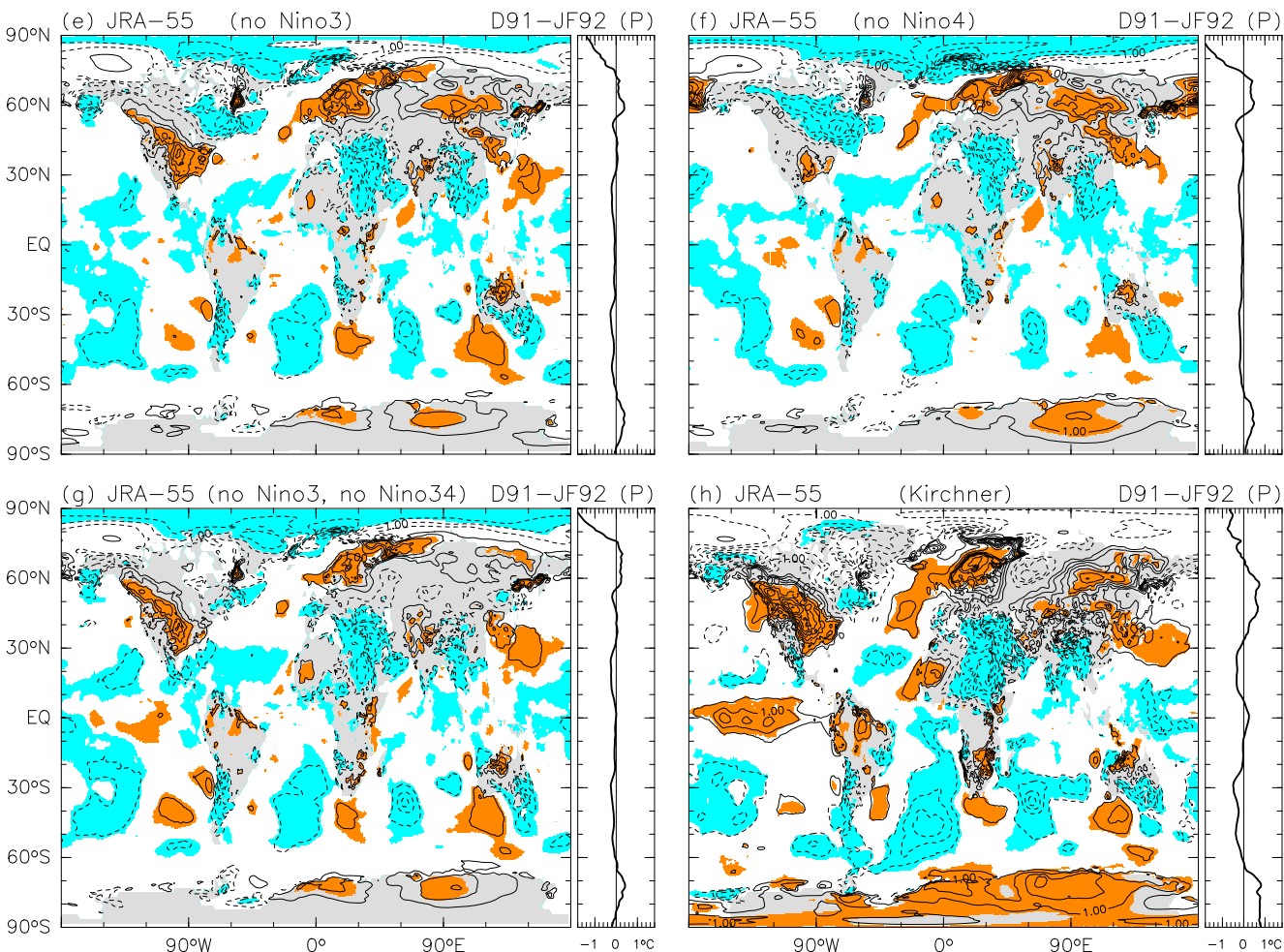

Figure B1. (continued)