# Peer review of "Surface temperature response to the major volcanic eruptions in multiple reanalysis data sets"

_Atmospheric Chemistry and Physics, 2019_

## Referee Comment (RC1) · Anonymous Referee #1 · 7 Jul 2019

General comments

The manuscript describes the response of the surface temperature to major volcanic eruption. The manuscript is in the scope of ACP. The authors applied multiple linear regression analysis (MLR) to retrieve the response from several atmospheric reanalysis products. Several studies of the climate response to volcanic eruptions based on model and observation results have been published before. The novelty of this work consists of the exploitation of multiple reanalysis data sets. However, the applications of similar observation data for all reanalysis products leads to visually similar, patchy and marginally significant response pattern making the manuscript difficult for the read-

ers. The problem is enhanced by a high number of very small panels and to simplified description of the figures. The manuscript is interesting and should be published, but the authors should make it more attractive for the readers. Otherwise, they great efforts will not be fully appreciated.

Major issues

1. I do not completely understand the motivation of the SST application as a proxy for MLR. Can be part of the signal lost due to this procedure.

2. Would it be possible to explain how the statistical significance was estimated? In text it is said that 1 SD criteria was used, but is it not too weak?

3. In the introduction I recommend describing expectations for the surface temperature response to volcanic eruptions. Then it will be easier to describe the obtained results.

Minor issues: 1. Page 8, first paragraph: How to explain similar anomalies during the periods w/o volcanos. Is it the results of similar observations used for assimilation?

2. Page 8, line 28: Why for two reanalyses only? Do the others produce similar results?

3. Page 10, first para: It is difficult to read. It looks like text description of the pattern visible from the figure

4. Page 11, line 1: More sophisticated methods are not always more accurate.

5. Page 11, second para: Not conclusive. What are the conclusions from the discussion?

6. Page 11: Too descriptive to my taste. I recommend improving the text.

7. Page 11, lines 24-26: Please, discuss why the difference between considered periods is so important.

8. Page 12, line 11: Maybe because it is not robust.

9. Page 12, line 22: "All known externally-forced". Could be internal than?

[Figure]
10. Page 12, lines 31-32: Not instructive conclusions. Kind a everything is possible...

11. Page 13, 3rd papa: Strong point. Can it be confirmed by radiative forcing consideration? How about models? They are not perfect, but at least does not depend on MLR problems.

---

## Referee Comment (RC2) · Fanglin Yang (Referee) · 8 Jul 2019

This study investigates changes of 2-m temperature over the globe following three major volcanic eruptions in the past few decades using 11 global atmospheric reanalysis data sets. Multiple linear regression (MLR) is used to remove variations of 2-m temperature that corresponds to or forced by seasonal harmonics, trends, QBO, solar cycle, and a combination of tropical SST modes. Then, residuals of the MLR is considered to be the signal of volcanic eruptions in the couple of years immediately following each of the three major eruptions.

Even though many investigations have been published in the past to understand the

impact of volcanic eruption on atmospheric circulation and surface temperature, it still presents a great challenge to quantify the impact with certainty. For observational study, it is difficult to separate changes in temperature induced by volcanic eruption from those induced by atmospheric and oceanic internal variability and external forcing. For numerical modeling, models may not be able to capture all natural variability, and the specification of volcanic forcing is often inaccurate. This study is based on linear regression and bears the same shortcomings of all statistical analyses; however, it is for the first time multiple analyses are used, and the residuals from different re-analysis data sets all showed similar patterns of cooling over the globe in the summer and fall following the three major volcanic eruptions. The authors also compared their MLR approach with the SVD approach found in previous studies, and confirmed both approaches produced similar cooling patterns. The magnitude of the cooling documented in this study is in general smaller than that reported in previous studies. This has implications for how to quantify volcanic forcing in numerical models for climate change study.

This manuscript is well written and well organized. I would recommend it be accepted for publication in ACP after the following few minor comments are addressed.

Minor comments 1) Please add a paragraph to the Introduction session to describe how volcanic eruption affects the surface temperature through direct radiative forcing and/or indirect changes in atmospheric circulation. 2) Is the 2m temperature response documented in this study consistent with the atmospheric temperature changes described in Fujiwara et al. (ACP, 2015) ? 3) It has been know that CFSR was constructed from a few different streams of analyses covering different time periods. Discontinuities are often found across the streams. Have the authors noticed the same feature and applied any technique to reduce the jumpiness?

---

## Author Comment (AC1) · 4 Sep 2019

**Response to Referee 1**

General comments

The manuscript describes the response of the surface temperature to major volcanic eruption. The manuscript is in the scope of ACP. The authors applied multiple linear regression analysis (MLR) to retrieve the response from several atmospheric reanalysis products. Several studies of the climate response to volcanic eruptions based on model and observation results have been published before. The novelty of this work consists of the exploitation of multiple reanalysis data sets. However, the applications of similar observation data for all reanalysis products leads to visually similar, patchy and marginally significant response pattern making the manuscript difficult for the readers. The problem is enhanced by a high number of very small panels and to simplified description of the figures. The manuscript is interesting and should be published, but the authors should make it more attractive for the readers. Otherwise, they great efforts will not be fully appreciated.

Thank you very much for your review. We made the panels for geographical distributions much larger for the open discussion, and believe that the current version of the panels available on the website is acceptable in terms of their size. We will also improve the description of the figures – please see below.

Major issues

1. I do not completely understand the motivation of the SST application as a proxy for MLR. Can be part of the signal lost due to this procedure.

The primary motivation of this approach is to remove the component of El Niño events right after the three volcanic eruptions as discussed in the first paragraph of the Introduction. It is common practice to use SST data to describe the El Niño Southern Oscillation (ENSO) time series, but there are actually several indices proposed/used as described in the fourth paragraph of the Introduction and in Table 2. The ENSO is an atmosphere-ocean coupling phenomenon affecting primarily the tropical Pacific. But, as again discussed in the fourth paragraph of the Introduction, there are similar basin-wide atmosphere-ocean coupling phenomena in the tropical Indian Ocean and in the tropical Atlantic Ocean. They are less impactful compared to the ENSO, and thus often neglected for upper air (e.g., stratospheric) studies. However, it is considered that they have a direct impact on the surface temperature over those oceans. Thus, we considered these as well.

Finally, some recent studies have revealed a wintertime teleconnection rooted in the Arctic Ocean that

influences East Asia and North America (as again described in the fourth paragraph of the Introduction.)

These are the reasons why we used SST data. In short: to remove those influences on surface air temperature that can be considered independent from the volcanic influence.

As discussed in detail in the final paragraph of Section 3 (Method), our approach is imperfect. A fraction of ENSO and other coupled atmosphere-ocean variability described using SST data may be inextricably linked to or even emerge from the volcanic response through forced changes in atmospheric circulation. Thus, the "volcanic response" reported in this paper should be regarded as the component of the volcanic forcing that is not mediated by coupled modes of atmosphere-ocean variability.

2. Would it be possible to explain how the statistical significance was estimated? In text it is said that 1 SD criteria was used, but is it not too weak?

We cannot use a rigorous statistical test for this study. Instead, we compare the residual during volcanic eruptions to its SD as a measure of how unusual the values are. This 1-SD criterion is used in previous studies on the surface temperature response to the Pinatubo eruption (e.g., Kirchner et al., 1999; Yang and Schlesinger, 2001).

3. In the introduction I recommend describing expectations for the surface temperature response to volcanic eruptions. Then it will be easier to describe the obtained results.

In the first paragraph of the Introduction, after the first two sentences (at page 2, line 3), we will add the following sentences:

The increased concentration of aerosols in the stratosphere causes a net negative radiative forcing at the surface (Robock, 2000), resulting in cold surface temperature anomalies when averaged globally or over the tropics. The geographical distribution of the surface temperature anomalies is, however, found to be much more complicated. Robock (2000) reviewed observations and theory of winter-time warming over the Northern Hemisphere (NH) continents (or the wave pattern of warm/cold anomalies) that result from changes in the tropospheric and stratospheric circulations after large eruptions. The surface temperature response at the regional scale is thus not only influenced by the direct radiative

forcing but also by the dynamical response of the atmospheric circulation. Studies on the geographical distribution of the surface volcanic response all show complex patterns of cooling and warming (e.g., Kirchner et al., 1999; Yang and Schlesinger, 2001).

Minor issues:

1. Page 8, first paragraph: How to explain similar anomalies during the periods w/o volcanos. Is it the results of similar observations used for assimilation?

For the case of the current study, we can say that very similar observations are assimilated for the full input reanalyses and for the surface input reanalyses, respectively. In this study, we do not attempt to explain the anomalies during the periods without volcanic eruptions.

2. Page 8, line 28: Why for two reanalyses only? Do the others produce similar results?

Yes, other reanalyses show similar results. We will add this information at page 9, line 19.

3. Page 10, first para: It is difficult to read. It looks like text description of the pattern visible from the figure

We described the features of the results, organizing the discussion similarly to previous figures; i.e., starting from the tropics, then NH extra-tropics, then SH extra-tropics, and then in and around Antarctica. These features are of course visible in the figures, but we think such a text description is necessary and useful for some readers.

4. Page 11, line 1: More sophisticated methods are not always more accurate.

Our method is more sophisticated in comparison to others by the removal of the influence of known forcings such as ENSO, Indian Ocean variability, Atlantic Ocean variability, etc. We do not claim it is more accurate.

5. Page 11, second para: Not conclusive. What are the conclusions from the discussion?

We will add the following sentence at the end of this paragraph:

Therefore, the cause of the transient cooling event in 1976 needs further investigation.

(When the ERA5 reanalysis is extended back to 1950 (most probably next year), we will re-investigate this issue.)

6. Page 11: Too descriptive to my taste. I recommend improving the text.

We assume that you are referring to the two paragraphs regarding Figures 6 and 7. They are indeed descriptive in nature. We have already made efforts to keep this part short, so that information is not duplicated and readers can proceed quickly to the next discussion.

7. Page 11, lines 24-26: Please, discuss why the difference between considered periods is so important.

We added this paragraph because this analysis provides a very good opportunity to test the robustness of the results. We will add the following sentence in the beginning of this paragraph:

The 1958–2001 analysis provides us not only the Mount Agung response but also an opportunity to test the robustness of the results following the eruptions of El Chichón and Mount Pinatubo.

8. Page 12, line 11: Maybe because it is not robust.

At least our results for the six reanalyses show very similar characteristics as described in the previous paragraph.

9. Page 12, line 22: "All known externally-forced". Could be internal than?

We will remove the word "externally" since it could be confusing. What is meant by the "forced component" is described later in the same sentence.

---

## Author Comment (AC2) · 4 Sep 2019

**Response to Referee 2 (Dr. Fanglin Yang)**

Thank you very much for your review.

This study investigates changes of 2-m temperature over the globe following three major volcanic eruptions in the past few decades using 11 global atmospheric reanalysis data sets. Multiple linear regression (MLR) is used to remove variations of 2-m temperature that corresponds to or forced by seasonal harmonics, trends, QBO, solar cycle, and a combination of tropical SST modes. Then, residuals of the MLR is considered to be the signal of volcanic eruptions in the couple of years immediately following each of the three major eruptions.

Even though many investigations have been published in the past to understand the impact of volcanic eruption on atmospheric circulation and surface temperature, it still presents a great challenge to quantify the impact with certainty. For observational study, it is difficult to separate changes in temperature induced by volcanic eruption from those induced by atmospheric and oceanic internal variability and external forcing. For numerical modeling, models may not be able to capture all natural variability, and the specification of volcanic forcing is often inaccurate. This study is based on linear regression and bears the same shortcomings of all statistical analyses; however, it is for the first time multiple analyses are used, and the residuals from different reanalysis data sets all showed similar patterns of cooling over the globe in the summer and fall following the three major volcanic eruptions. The authors also compared their MLR approach with the SVD approach found in previous studies, and confirmed both approaches produced similar cooling patterns. The magnitude of the cooling documented in this study is in general smaller than that reported in previous studies. This has implications for how to quantify volcanic forcing in numerical models for climate change study.

This manuscript is well written and well organized. I would recommend it be accepted for publication in ACP after the following few minor comments are addressed.

Thank you very much for your evaluation.

Minor comments

1) Please add a paragraph to the Introduction session to describe how volcanic eruption affects the surface temperature through direct radiative forcing and/or indirect changes in atmospheric circulation.

In the first paragraph of the Introduction, after the first two sentences (at page 2, line 3), we will add

the following sentences:

The increased concentration of aerosols in the stratosphere causes a net negative radiative forcing at the surface (Robock, 2000), resulting in cold surface temperature anomalies when averaged globally or over the tropics. The geographical distribution of the surface temperature anomalies is, however, found to be much more complicated. Robock (2000) reviewed observations and theory of winter-time warming over the Northern Hemisphere (NH) continents (or the wave pattern of warm/cold anomalies) that result from changes in the tropospheric and stratospheric circulations after large eruptions. The surface temperature response at the regional scale is thus not only influenced by the direct radiative forcing but also by the dynamical response of the atmospheric circulation. Studies on the geographical distribution of the surface volcanic response all show complex patterns of cooling and warming (e.g., Kirchner et al., 1999; Yang and Schlesinger, 2001).

2) Is the 2m temperature response documented in this study consistent with the atmospheric temperature changes described in Fujiwara et al. (ACP, 2015) ?

It is not easy to compare because Fujiwara et al. (2015) looked at 1-year averaged responses, while the current study examines 3-month averaged responses. Also, Fujiwara et al. (2015) used pressure-level data, where temperatures are extrapolated down to 1000 hPa in most reanalyses over land (except for MERRA). Despite these differences, the 1000 hPa level results in Fujiwara et al. (2015) and the zonal mean surface results in the current study are consistent with each other in the sense that they both show qualitatively similar cooling responses.

3) It has been know that CFSR was constructed from a few different streams of analyses covering different time periods. Discontinuities are often found across the streams. Have the authors noticed the same feature and applied any technique to reduce the jumpiness?

All other reanalyses also have execution streams (e.g., Figure 2 of Fujiwara et al., 2017). We have not applied any special treatment for the stream change points. We have not noticed any issues that may be related to the stream changes in this volcanic study.

Fujiwara, Wright, et al. (2017), Introduction to the SPARC Reanalysis Intercomparison Project (S-RIP) and overview of the reanalysis systems, Atmospheric Chemistry and Physics, 17, 1417-1452, doi: 10.5194/acp-17-1417-2017.

---

## Author Response (AR2)

**Dear Editor Dr. G. Stiller:**

**In this letter, we include**
- **detailed point-by-point response to the comments by Referee 3**
- **a changes-tracked version of the text**

**Thank you very much for your consideration.**

**Masatomo Fujiwara (on behalf of all coauthors)**

The authors apply multi-linear regression techniques to identify responses in near-surface air temperature to three large volcanic eruptions that occurred during the 20th (Agung, El Chichon, Pinatubo) in several reanalysis data-sets. The topic is interesting and certainly very appropriate for ACP. To my understanding there are two major results, a) anomalies after large eruptions are very similarly represented in all analyzed data sets, b) the anomalies are patchy and globally relatively small compared to earlier analyses, e.g. the authors suggest that the maximum response to the Pinatubo eruption is in the order of 0.1 to 0.15 K when averaging over 60S to 60N. While I think that result a) is convincingly presented and useful, the authors failed to convince me that their technique provides a more reliable estimate for the responses than earlier studies. I would suggest publication only after a major revision discussing the results much more carefully in comparison to earlier studies. Furthermore, I think that while the paper is overall very clearly written, most of the figures could be substantially improved to better support the points that the authors would like to make.

Thank you very much for your review. In the revised manuscript, we place greater emphasis on the limitations of our method. In this response letter, we present an alternative set of geographical figures (i.e. a reference distribution and the difference of each reanalysis/method from that reference) and explain why we have decided to show "absolute" fields in the manuscript. Please see below for details.

Concerning the comparison to earlier studies I appreciate that the authors compare several different analysis methods in appendix A. I think it is a very important conclusion that "differences among the different methods are generally much greater than the differences among different reanalysis data sets". But I don't think the appendices provide convincing support for the authors believe that their estimates are "more appropriate because [they] have more thoroughly considered potential confounding factors outside of the volcanic eruptions themselves". Indeed, with their technique the authors include 11 SST indices for which they make an effort to be orthogonal. However, a general issue is that the volcanic eruptions should have an impact on SSTs themselves and the larger the number of indices in the MLR, the more likely a projection of volcanic signals onto them seems to me. In addition, there is some literature discussing the possibility of an effect of volcanic eruptions on ENSO, which, if it exists, would make it necessary to question the inclusion of ENSO indices in such an MLR at all. The authors admit the problem early in the text, saying that their method "assumes that the zonally-symmetric volcanic aerosol forcing does not project substantially onto strongly asymmetric modes of variability like ENSO". But they don't discuss later on, in particular not in the conclusions, how this would affect their results, and how they come to their earlier mentioned "believe" despite these issues. One possibility to make a step into the direction of further evaluating their method would be to leave out

the volcanic periods in the MLR. But even with such a test I don't see that the issue of a clear identification of the volcanic signals can be solved and a fair comparison of different methods can be reached using only reanalysis data sets. So I would rather suggest that the authors openly discuss the existing problems than to state believes. This should start already in the introduction where several earlier studies have been mentioned that have analyzed volcanic signals, but it is not made clear what potential weaknesses of these studies were and how they could be overcome. I also would like to see this discussed in the conclusions: How to make further progress concerning this issue that would go beyond believes.

Following these comments and suggestions, we have made the following revisions.

(1.a) In Section 1 Introduction, we had already discussed potential weaknesses of past studies (in the second paragraph), including not explicitly subtracting the ENSO component for most studies, not investigating the impact of each eruption separately in recent studies, and not using and comparing several reanalysis products. At the end of this paragraph, we write, "The current study aims to expand on these previous works by investigating the surface temperature response both in latitudinal means and in geographical distribution, treating each of the three major eruptions separately, explicitly subtracting ENSO and other known forced components, and using global atmospheric reanalysis data." However, we agree that we had not discussed the limitations of our method in the Introduction, deferring these to the end of Section 3 ('Method') in the previous manuscript. Thus, we have moved that paragraph to the Introduction, with some changes, and it is now the second last paragraph in the revised version. In the following, we copy the moved paragraph:

"We note that our approach to isolate the surface temperature anomalies associated with volcanic eruptions from other external forcings is imperfect. One important limitation is that a fraction of ENSO-related variability may emerge from the volcanic response through forced changes in atmospheric circulation (e.g., Wang et al., 2018, and references therein). Our method implicitly assumes that the zonally-symmetric volcanic aerosol forcing does not project substantially onto strongly asymmetric modes of variability like ENSO. However, the impacts of volcanic eruptions on ENSO-related variability and other modes of coupled atmosphere–ocean variability are not well characterized and thus some uncertainties related to this influence remain in our analysis. The temperature anomalies following volcanic eruptions as reported below should be regarded as the component of the volcanic forcing that is not mediated by coupled modes of atmosphere–ocean variability."

Wang, T., Guo, D., Gao, Y., Wang, H., Zheng, F., Zhu, Y., Miao, J., and Hu, Y.: Modulation of ENSO

evolution by strong tropical volcanic eruptions, Clim. Dyn., 51, 2433–2453, https://doi.org/10.1007/s00382-017-4021-2, 2018.

(1.b) In Section 5 Conclusions, we agree that we did not explicitly discuss the limitations of our method. Also, we have used an inappropriate word "believe" in the fourth paragraph. Thus, we have made the following revisions:

(1.b.1) The following sentences have been added to the end of the first paragraph of Section 5 Conclusions:

"As discussed in the Introduction, our method has limitations. In particular, our method neglects possible responses of ENSO or other major modes of internal variability to the eruptions, which may cause our residual term to underestimate the scale of the response."

 (1.b.2) The fourth paragraph has been replaced with the following one:

"In comparison with previous studies, our zonal-mean results tend to imply smaller cooling magnitudes following the major volcanic eruptions. We have more thoroughly considered potential confounding factors outside of the volcanic eruptions themselves. The anomalies may be underestimated by our method if the volcanic eruptions analyzed in this study directly influenced ENSO variability (e.g., Wang et al., 2018, and references therein). However, at the very least, we argue that a global cooling of ~0.5 K as claimed by studies from the 1990s (e.g. Hansen et al., 1992; Parker et al., 1996) and referred to in more recent discussions on geoengineering (e.g. Crutzen, 2006) overestimates the actual response to the Pinatubo eruption. This contention is also supported by Figures 9.8 and 10.6 of IPCC (2013) (see also discussion in the second paragraph of section 4.1 above). More appropriate values may be closer to our results, i.e. 0.10–0.15 K for the 60°N–60°S mean, although including polar regions would amplify the uncertainty in this estimate."

A further issue I have concerns the presentation of spatial patterns of anomalies after different volcanic eruptions. Of course, one can't expect the responses to be the same because of the different characteristics of the different eruptions. However, any comparison and possibility to identify common signals is made impossible by the choice to show anomalies for different seasons. I understand that the authors want to present periods with the largest anomalies, but I'd ask them to try analyzing the same seasons and depending on the outcome show these results or just make a statement if there aren't any similarities.

Based on the residual time series in Figures 1 and 5, we decided to focus on the periods with peak cooling. Because the eruptions occurred at different months of the year, looking at all the responses at the same season means looking at different stages of the evolution. In practice, the peak cooling for the Pinatubo eruption occurred in SON, while that for the other two eruptions occurred in JJA. Thus we prepared Figure 3 to show JJA anomalies based on JRA-55 and R-1 for the Pinatubo case. The JJA anomaly is different for different eruptions, in part because the stratospheric aerosol loading was very different among the three eruptions (see Figure 9c of Fujiwara et al., 2015, https://doi.org/10.5194/acp-15-13507-2015).

To be honest, I don't expect similar patterns because my guess is that even for similar eruption characteristics, patterns of single events may look very different because they are not dominated by the response to volcanoes but internal variability. This could be tested, e.g. by comparing the residuals after the eruptions to residuals outside of volcanic episodes. If indeed responses may not be dominant, I'd also ask the authors to reconsider their use of language, which is unfortunately not consistent. Sometimes they are careful in their wording, but more frequently they are talking about "responses to volcanic eruptions" while in some places they mention that the patterns would include both, responses (with all the uncertainty mentioned above) and internal variability.

Again, we would like to point out that the stratospheric aerosol loading was very different, in both the column amount and latitudinal distribution, among the three eruptions.

Regarding the terminology, in Section 4 Results and Discussion, in Section 5 Conclusions where applicable, and in the Appendices, we have changed the term "response" to "anomaly" because we agree that the term "response" may be too strong, considering the uncertainties in our method, and because many previous studies often used the term "anomaly."

My final major point concerns the figures containing spatial patterns, of which I find the isolines very hard to read. E.g., to make the important Fig. 9 easier to read I think there are better ways than just to use green color above some threshold. In general I'd ask the authors to present clearly the information they want to convey to the reader and simply not show what they consider unimportant. When it comes to showing differences between different methods or data sets, it may be more informative to actually show difference plots than absolute fields.

Figure 9 has been revised, with three different shades of color:

[Figure]

**Figure 9, revised.**

In the following pages, we show alternatives to Figures 2, 4, 6, 7, 8, A1, A2, and B1; i.e., the reanalysis ensemble mean (REM) and the difference of each reanalysis from the REM for Figures 2, 4, 6, 7, and 8, and the difference of each method from the main method for Figures A1, A2, and B1. The alternative figures for the main body of the paper essentially confirm what we show in Figure 9: differences are largest in the polar regions; differences are large over the continents; and older reanalyses and surface input reanalyses tend to show larger differences over the continents. However, it is important to note that there is no guarantee that the REM is the best estimate. The alternative figures for the Appendices are much more complicated and less suitable for illustrating the key points discussed in the text.

We think that both approaches have advantages and inconveniences. Showing the "absolute" fields is good in that it allows to compare easily the spatial pattern of the responses; we agree that it is a bit harder to judge how a specific reanalysis compares to others, but still possible. Showing the "difference" fields gives a precise deviation of a single reanalysis from a reference, but it becomes much harder to assess how well the reanalyses agree on the broad features of the cooling and warming patterns. Furthermore, in some regions, we may question the validity of the REM as a reference. Considering these factors, we choose to show "absolute" fields in the manuscript.

One final minor comment concerns the very last statement in the conclusion section. Of course, one needs to carefully analyze potential signals of SRM, but when it comes to modelling one can easily avoid complications discussed here in the context of the single realization of reality.

We have removed "modelling" and rephrased the final sentence as:

"Thus, evaluating the effects of SRM, if implemented, would not be an easy task in the real climate system."

The alternative Figures to Figures 2, 4, 6, 7, 8, A1, A2, and B1 are shown in the following pages. Please also note that a changes-tracked version of the manuscript is attached to the end of this letter.

[Figure]

**Figure R1. Geographical and zonal-mean distributions of the 2-metre temperature anomalies averaged from September to November 1992 following the Mount Pinatubo eruption that occurred June 1991 using the Reanalysis Ensemble Mean (REM) which includes the 10 reanalysis data sets shown in Figure 2.**

[Figure]

**Figure R2. Geographical and zonal-mean distributions of the difference from the REM (shown in Figure R1) for each of the 10 reanalysis data sets shown in Figure 2.**

[Figure]

**Figure R2. (continued)**

[Figure]

**Figure R3. Geographical and zonal-mean distributions of the 2-metre temperature anomalies averaged from June to August 1983 following the El Chichón eruption in April 1982 using the Reanalysis Ensemble Mean (REM) which includes the 10 reanalysis data sets shown in Figure 4.**

[Figure]

**Figure R4. Geographical and zonal-mean distributions of the difference from the REM (shown in Figure R3) for each of the 10 reanalysis data sets shown in Figure 4.**

[Figure]

**Figure R4. (continued)**

[Figure]

**Figure R5. Geographical and zonal-mean distributions of the 2-metre temperature anomalies averaged from September to November 1992 following the Mount Pinatubo eruption in June 1991 using the Reanalysis Ensemble Mean (REM) which includes the 6 reanalysis data sets shown in Figure 6.**

[Figure]

**Figure R6. Geographical and zonal-mean distributions of the difference from the REM (shown in Figure R5) for each of the 6 reanalysis data sets shown in Figure 6.**

[Figure]

**Figure R7. Geographical and zonal-mean distributions of the 2-metre temperature anomalies averaged from June to August 1983 following the El Chichón eruption in April 1982 using the Reanalysis Ensemble Mean (REM) which includes the 6 reanalysis data sets shown in Figure 7.**

[Figure]

**Figure R8. Geographical and zonal-mean distributions of the difference from the REM (shown in Figure R7) for each of the 6 reanalysis data sets shown in Figure 7.**

[Figure]

**Figure R9. Geographical and zonal-mean distributions of the 2-metre temperature anomalies averaged from June to August 1964 following the Mount Agung eruption in March 1963 using the Reanalysis Ensemble Mean (REM) which includes the 6 reanalysis data sets shown in Figure 8.**

[Figure]

**Figure R10. Geographical and zonal-mean distributions of the difference from the REM (shown in Figure R9) for each of the 6 reanalysis data sets shown in Figure 8.**

[Figure]

**Figure R11. An alternative of Figure A1. Panels (b)–(e) have been replaced with the difference from Panel (a).**

[Figure]

**Figure R12. An alternative of Figure A2. Panels (b)–(e) have been replaced with the difference from Panel (a).**

[Figure]

**Figure R12. An alternative of Figure B1. Panels (b)–(g) have been replaced with the difference from Panel (a). Panel (h) (based on the method by Kirchner et al., 1999) has been omitted.**

[Figure]

**Figure R12. (continued)**

[revised manuscript text omitted]